# PREPARE YOUR VIDEO FOR STREAMING WITH SEGUE

Melissa Licciardello    Lukas Humbel    Fabian Rohr    Maximilian Grüner    *Ankit Singla*
*ETH Zürich*            *ETH Zürich*    *ETH Zürich*    *ETH Zürich*          *ETH Zürich*

## Abstract

We identify new opportunities in video streaming, involving the joint consideration of offline video chunking and online rate adaptation. Due to a video's complexity varying over time, certain parts are more likely to cause performance impairments during playback with a particular rate adaptation algorithm. To address such an issue, we propose SEGUE, which carefully uses variable-length video segments, and augment specific segments with additional bitrate tracks. The key novelty of our approach is in making such decisions based on the video's time-varying complexity and the expected rate adaptation behavior over time. We propose and implement several methods for such adaptation-aware chunking. Our results show that SEGUE substantially reduces rebuffering and quality fluctuations, while maintaining video quality delivered; SEGUE improves QoE by 9% on average, and by 22% in low-bandwidth conditions. Finally, we view our problem framing as a first step in a new thread on algorithmic and design innovation in video streaming, and leave the reader with several interesting open questions.

## 1  Introduction

Video-on-demand adaptive bitrate streaming (ABR) requires the video provider to partition their video content into segments of a few seconds worth of playback time, and encode each segment at multiple quality levels. This allows clients with time-variable network conditions to adaptively choose video quality. The video quality decisions are made by a rate adaptation algorithm, with the goal of improving the client's quality of experience (QoE) by delivering the highest quality video, without pauses and infrequent quality switching.

ABR adaptation algorithms are well-studied, with many recent high-quality proposals [4, 18, 26, 37]. The problem framing in this literature is that of using a provider's given QoE function to construct an ABR algorithm that will result in high QoE, as much as possible, when operating online across a range of network conditions and video content.

However, this framing leaves out the offline, provider-side phase of ABR: video chunking. We use the term "chunking" to refer to cutting a video into segments, and determining what set of bitrates each segment will be encoded in. Most prior works and deployed streaming platforms use constant-length segmentation, typically 4-6 seconds, and the same number of bitrate tracks across each video segment within one video. While some prior work (§2.2) has explored relaxing these

constraints, we posit that there are new and unexplored opportunities at the intersection of the offline and online phases of ABR streaming. Specifically, offline chunking can be tuned based on the expected playback behavior of a video under a provider's online adaptation algorithm.

Video complexity varies over time. Indeed, we observe that some parts of a video are likelier to suffer from performance impairments such as lower quality, rebuffering, and frequent bitrate track switches during playback. Even two similarly complex segments of a video may differ in their vulnerability to performance impairments due to their surrounding context, *e.g.,* a complex segment preceded by many low-complexity segments differs from one preceded by many high-complexity ones: These two different scenarios might highly impact the buffer health of the player and, as a consequence, the rebuffering probability. Finally, using different rate adaptation algorithms for the same video and network conditions can also change the same segment's vulnerability to impairments.

While prior work has explored time and space variability on a per segment granularity, to the best of out knowledge there is no prior work that accounts for playback context dependence and adaptation algorithm dependence in tuning video chunking. With SEGUE, we account for these factors in exploring the tuning of chunking along two axes: (a) segmentation, *i.e.,* deciding the lengths and boundaries of video segments that a client can fetch; and (b) augmentation, *i.e.,* adding to the provider's current bitrate track design, additional bitrate tracks for a small fraction of segments, such that more bitrate options are available to online adaptation for these segments.

SEGUE uses a simulation-based method, exploring and evaluating segmentations and augmentations of a video across a broad set of network traces. In such simulations, the provider's ABR adaptation algorithm is used to make decisions, and their QoE function is used rank the candidate chunkings. We compare SEGUE's chunking performance to several heuristics drawn from intuition, revealing how the inability of the latter to account for context and adaptation leads to significantly worse performance than our proposal. While some of these heuristics still improve over constant-length segmentation and a constant set of bitrate tracks, the improvements are just smaller than SEGUE's.

We implement SEGUE atop an unmodified H.264 video encoding pipeline. Our implementation uses *ffmpeg* with the *libx264* library. We modify the reference DASH player im-

plementation [31] to support SEGUE, but like past work [18], use this implementation only to demonstrate the high fidelity of a simulator that is orders of magnitude faster. We then use the simulator to extensively evaluate the performance with SEGUE's chunkings across a diverse set of videos and network traces, and four adaptation algorithms from past work.

We show that especially in low-bandwidth conditions, SEGUE yields large improvements in QoE, 9% on average across traces and videos (§7.7).

To summarize, we make the following contributions:

- We propose to optimize offline chunking, considering both the playback context and the adaptation algorithm.

- In this framework, we explore various methods for *segmentation* of a video into variable-length segments.

- We explore the *augmentation* of parts of a video with additional bitrate tracks to help adaptation make finer-grained decisions and improve QoE.

- We evaluate SEGUE extensively, showing how its improvements depend on video content and adaptation algorithms. We also comment on the provider-side costs of SEGUE.

Perhaps even more valuable than SEGUE's optimization methods and results, are the questions it sets up for future work, on how we might co-design the offline and online phases on ABR streaming. For the benefit of future research along this path, we release SEGUE's implementation, together with our high-fidelity simulator [14].

## 2    Background and related work

### 2.1    Video streaming 101

Adaptive bitrate video streaming comprises two pieces, video encoding, which runs offline at the content provider, and video adaptation, which runs online, typically at the client. Together, these optimize for improving quality of experience for clients, while limiting resource usage for the provider.

**Encoding:** Offline, a video is encoded into multiple tracks, each of a different *bitrate*. The bitrate describes compression, *i.e.,* the bits per second used to encode the content. Different tracks are described by their average bitrate, with substantial variation around this average due to variable bitrate encoding; this allows complex scenes to benefit from a higher than average bitrate, while reducing bitrate for simple scenes. The bitrates of different tracks are chosen for different target viewing *resolutions*. If a bitrate targeted at a lower resolution is delivered to a higher-resolution client viewing screen, the video can be scaled up, with some "pixelation". Video may be encoded such that for the same target resolution, multiple tracks with different bitrates are available.

Typically, each track is broken into fixed-length *segments*. For continuity when playback switches from one track to another, this segmentation must meet two constraints: (**C1**) segment boundaries of different tracks must be aligned in

terms of content; and (**C2**) each segment starts with a *key frame*, *i.e.,* one encoded without reference to previous frames.

**Adaptation:** Online, an adaptive bitrate adaptation algorithm decides on which video bitrate-track to use. If the network provides consistently high bandwidth, the highest-bitrate track that makes a perceivable difference for a particular screen size can be used; otherwise, as bandwidth varies over time, lower-bitrate tracks may be used dynamically. A client-side buffer is used to absorb some bandwidth variability by storing video segments for future playback, but large or persistent bandwidth changes require shifting to a lower-bitrate track; otherwise, the playback buffer will empty out, and the client will see a pause or *rebuffer*.

**Client QoE:** Quality of experience metrics assess viewer satisfaction with video streaming. The relevant metrics include:

- The sum of bitrates across segments played;

- The sum of pause or rebuffer times;

- The sum of bitrate differences from track switching.

Typically, a weighted sum of these metrics is used as a QoE function, with the weights drawn from past measurement work [37]. In line with newer work driven by industry shifts, instead of just bitrate, we use Netflix's VMAF score for perceptual quality [13, 20], which uses a learning model to assign a score to a segment's playback at a certain bitrate in line with what a human would rate its quality as on a certain screen size. The raw video has VMAF, $\mathcal{V}_{\mathrm{raw}} = 100$, with different bitrates leading to VMAF scores from 0 to 100. We use the VMAF mobile, HDTV, and 4K models.

If $\mathcal{V}_i$ is the VMAF of the $i^{\mathrm{th}}$ segment at the track used for it, and $R_i$ is the rebuffering time incurred during the $i^{\mathrm{th}}$ segment being played, then for a video with $N$ segments in all, QoE is calculated as:

$$QoE = \lambda \sum_{i=0}^{N} \mathcal{V}_i - \beta \sum_{i=0}^{N} R_i - \gamma \sum_{i=1}^{N} |\mathcal{V}_i - \mathcal{V}_{i-1}|. \qquad (1)$$

$\lambda$, $\beta$, and $\gamma$ are weights reflecting the value of each metric.

**Provider resource usage:** While client QoE is the determinant for many provider decisions, providers also want to contain infrastructure costs. Encoding video is compute intensive, and storing the segments for a large number of different tracks consumes storage at distributed caches. Thus, providers also attempt to limit the complexity of the encoding pipeline, and the number of tracks per video.

### 2.2    Related work

Adaptive bitrate video streaming is a broad research area; we only discuss the work closest to SEGUE's ideas.

**Industry efforts:** Netflix's per-scene encoding [22] exploits the relative homogeneity of content comprising one scene within a video, to refine encoding decisions. The outcome of this process is *still* a fixed number of tracks per video. In

a subsequent blog [23] Netflix lays out how to merge shots into streamable segments, using a strategy that is very close to SEGUE's *Time* heuristic, explained in §4.

The innovation is rather in what specific bitrate is being used at each encoding point. Per-scene encoding, by selecting appropriate bitrates for each scene on each track, could potentially increase bitrate variations across scene boundaries, and thus provide *more* opportunities for SEGUE's methods. Absent an available implementation of Netflix's ideas, we have not quantified the impact of this yet.

Measurement work on YouTube [19] observed variable-length segments, with some evidence that during adaptation, YouTube uses shorter segments in response to bandwidth fluctuations. Unfortunately, no details are publicly known on how these are encoded or used. For instance: is each video coded with multiple equal-length segmentations? If so, how are the lengths decided, and how many different lengths are encoded? Alternatively, if segment lengths are indeed non-uniform like SEGUE, with only some parts of a video available in shorter segments, how are these parts chosen? Even if YouTube *is* pursuing a SEGUE-like method, that would only underscore the value in an *open* investigation of these ideas.

**Segmentation:** Prior work [9, 35, 38] has explored aligning scene boundaries and segmentation to improve coding efficiency by grouping homogeneous content together. As noted above for per-scene encoding, SEGUE's ideas are orthogonal to this, and address grouping which scene fragments into segments will result in beneficial rate adaptation behavior. Other work has attempted to optimize the (fixed) size of segments with the goal of improving transport [16] or HTTP protocol efficiency [15]. SEGUE's constraints like avoiding "too small" segments also address some of these problems, but its primary objectives and methods are very different: producing a variable-length segmentation that results in desirable rate adaptation behavior and high QoE.

The closest prior work [30] simply uses video group of picuters (GOP) as segments. This approach is prone to performance pitfalls, as discussed in §7.5.1. Another prior effort [34] suggests segmenting the same video multiple times with different (fixed) segment lengths, to allow the client greater flexibility during adaptation. SEGUE's approach naturally inherits this property when it is desirable, without the expense of multiple redundant segmentations, as discussed in §3.2. SEGUE also allows more flexibility by not being restricted to a small set of fixed-length segmentations, allowing natural keyframe boundaries to determine segment length.

**Augmentation**: The closest prior works [27, 28] pursue the opposite of SEGUE's strategy, *i.e.,* removing redundant segments to reduce storage costs or to improve bandwidth utilization. For instance, in Fig. 2, few segments across different tracks encode a near-identical perceptual quality; one could keep only the lower bitrate version, removing the higher bitrate ones, without much performance impact.

In contrast, SEGUE's optimization for improvements in QoE requires a completely different methodology, where accounting for playback context and rate adaptation algorithm is useful, as we show later. In §7.5.2 we compare the performance of [27, 28] to SEGUE's approach, and we elaborate on how we could merge them, in order to both account for playback context and optimize for bandwidth utilization and storage.

## 3 New opportunities in streaming

We draw out SEGUE's motivating observations using a running example of a video encoded using H.264 with variable bitrate encoding. The video is encoded into constant-length segments of 5 seconds each, across multiple bitrate tracks. (The details of the encoding are left to §5.) Using two rate adaptation algorithms, we evaluate the streaming behavior aggregated across a large set of traces. To avoid the effects of startup behavior, we show results starting at the 25th segment, *i.e.,* 125 seconds into playback.

Fig. 1(a) shows the variability of the video bitrate across segments for 3 tracks. As expected, variations for different tracks are in close alignment. Segment S37, S35, and S30 are the most complex, with the encoder using the highest bitrates, while S29, S33, and S39 are the simplest segments.

We partition our traces into bandwidth buckets, each bucket containing traces with average-over-time bandwidth in a certain range: 0.5-1 Mbps, 1-1.5 Mbps, 1.5-2 Mbps, etc. Details concerning the utilized trace sets can be found in §6.

We then tested the behaviour of two adaptation algorithms, a rate-based (RB) and a buffer-based (BB) (which are described in §5). For both rate adaptation algorithms, we compute the (observed) probability of rebuffering at any point in playback across traces in each bucket. Fig. 1(b) shows the probability distributions for three trace buckets (S – slow, M – medium, F – Fast) for a buffer-based (BB) adaptation algorithm from past work. Fig. 1(c) shows the corresponding average seconds of playback available in the client's buffer.

**Playback context dependence:** We observe from Fig. 1(a) and (b), that the probability of rebuffering of segments of similar complexity (bitrate) differs substantially depending on their placement in the stream. For instance, particularly for the lower-bandwidth bucket, even though S30 has lower bitrate than S37, S30 is substantially likelier to incur rebuffering. Thus, the playback context of a segment impacts its likelihood of suffering from performance impairments.

**Adaptation algorithm dependence:** Fig. 1(d) shows the average buffer occupancy for a rate-based adaptation algorithm, showing the stark contrast with BB in Fig. 1(c). Due to an higher buffer occupancy, we observe that the RB algorithm is less likely to incur rebuffering events (the graph has been omitted for readability). Thus, for the same video and network traces, the same segment's vulnerability to performance impairments depends on the adaptation algorithm in use. (This is indeed obvious, our contribution is in accounting for and

using this dependence.)

**Network trace dependence:** While ABR algorithms handle instantaneous bandwidth fluctuations, SEGUE finds common patterns among streaming sessions which lead to the determination of vulnerable parts of the video. These vulnerabilities depend not only on the expected playback state but also on the ABR adaptation logic. SEGUE uses a large number traces to minimize the effect of a single trace's fluctuation. The different trace sets are discussed in §6.

### 3.1 What levers can we tune?

Online rate adaptation must cope with highly unpredictable network bandwidth changes. However, the other time-varying determining factor, *i.e.,* video complexity variation and its interaction with rate adaptation, is more predictable, and can be accounted for in offline video chunking. We use two levers to adjust chunking to this end:

- Segmentation: we can adjust the lengths and boundaries of the video segments a client can later fetch.

- Augmentation: for select segments, we can add bitrate tracks to provide greater flexibility to online adaptation.

We next describe why these levers are interesting to tune, and some intuitions on how this might be done.

**Segmentation:** Fig. 2 shows the instantaneous bitrate per frame over playback time. The video is encoded using *ffmpeg* and H.264, with two-pass encoding. The red dashed lines show the key frames, with the rest of the frames encoded with reference to these. The maximum interval between successive key frames is passed as an argument to the encoder, and is 5 seconds in this instance. As Fig. 2(a) shows, key frames are not distributed uniformly across time: typically, relatively static parts of a video will feature larger gaps between successive key frames, while in complex, motion rich parts, key frames will occur more frequently. This property enables the use of key frames to group parts of the video with similar characteristics together. Recall constraint C2 from §2.1: video segments must each start with a key frame. We can thus collect such sets of adjacent key frames to form segments, but this will result in segments of non-uniform length. In contrast, a fixed-segment length setup forces the encoder to add key frames at fixed intervals corresponding to segment length, with additional key frames within each segment, as necessary. By carefully shaping segments of non-uniform length, we can let the client fetch shorter segments during parts of a video more vulnerable to streaming impairments, thus allowing finer-grained rate adaptation decisions. For less vulnerable parts, longer segments can be used.

Another aspect where variable-length segments help relates to the stability of perceived playback quality. Fig. 2(b) shows the perceptual quality (VMAF) computed *per frame* for a few constant-length segments of an action movie. For the segments highlighted in pink, there is a huge fluctuation in VMAF within the segment boundaries, perhaps due to a

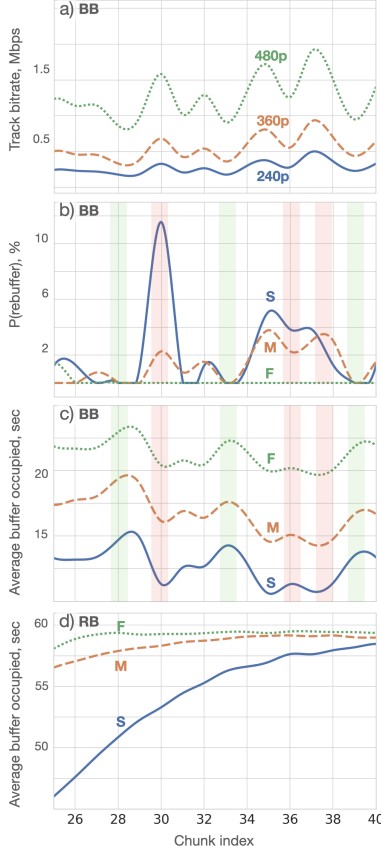

**Fig. 1:** *The complexity of video content varies over time, and its interaction with the used rate adaptation algorithm determines which segments are most vulnerable to playback impairments. Fig. (a) shows the variability in bitrate for three different resolutions (480p, 360p and 240p). Fig. (b) shows the observed probability of rebuffering for the segments plotted in Fig. (a) for a buffer-based algorithm for three different traces buckets (blue line: Slow, orange line: Medium, green line: Fast). Fig. (c) shows the correspondent average buffer occupancy. We highlight that higher observed rebuffering probability correspond to drops into the average buffer occupancy (red shaded in the picture). Conversely, drops into the observed rebuffering probability correspond to higher average buffer occupancy. Fig. (d) plots the average buffer occupancy for the same segments for a rate-based algorithm for the same traces buckets. The buffer behaviour varies substantially with respect to a buffer-based algorithm.*

scene change. There can also be value in synchronizing these change points with segmentation, allowing more informed adaptation decisions that account for such changes.

**Augmentation:** Fig. 2(c) shows another aspect of temporal variability using a video segmented into 5-second segments, with the average bitrate of each segment plotted across 4 tracks. Due to variable bitrate coding, the per-segment bitrate within each track varies substantially. In particular, the segment from 15–20 sec uses a much higher bitrate, so much so, that its bitrate at any track is comparable to the the rest of the video's bitrate at one higher track. This is because the encoder decides that the scenes of high complexity in this

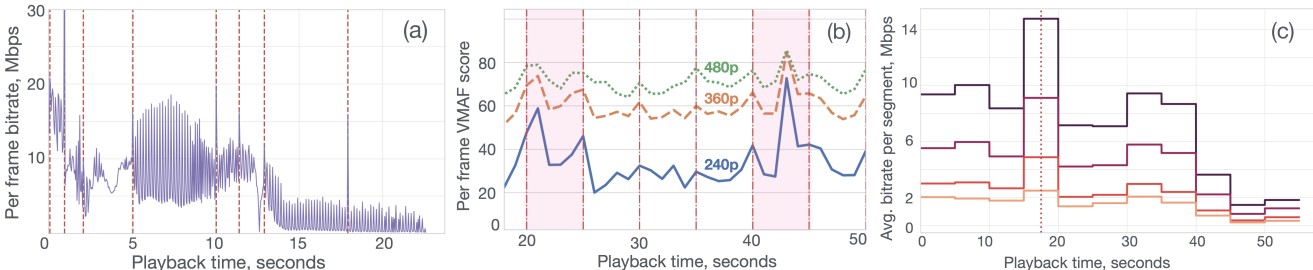

**Fig. 2:** *(a) The violet (solid) line is the bitrate per frame, while the red (dashed) line marks the keyframes. (b) Breaking the video into fixed length segments produce segments with high internal perceptual quality instability. (c) Average bitrate for a video encoded at 4 resolutions: due to VBR encoding, the bitrate per segment varies.*

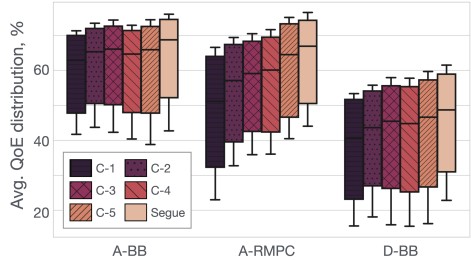

**Fig. 3:** *The distribution of QoE per second of playback (normalized to max. possible QoE) across our test network traces for video **A** with BB and RMPC rate adaptation, and video **D** with BB adaptation. The box-plot shows the quartiles, with whiskers for $20^{th}$ and $80^{th}$ percentiles.*

duration warrant higher bitrate for sufficient video quality.

Unfortunately, even with the freedom of variable bitrate coding, it is sometimes either hard to achieve sufficient perceptual quality for complex segments, or higher-than-necessary bitrate is "wasted" on simple segments. For this problem, and the bitrate peaks illustrated in Fig. 2(c), instead of using the same number of tracks throughout, we could augment the encoding of segments vulnerable to streaming impairments with more tracks. These added choices would enable more fine-grained decisions during adaptation.

### 3.2 The need of variability
Before delving into SEGUE's design, we first illustrate the performance problems with baselines that do not account for variability in segments length and number of tracks.

We evaluate the QoE with different constant-length segmentations, ranging from 1 to 5 seconds (C1, C2, …, C5). Fig. 3 shows the QoE achieved across a set of test network traces for video **A**, using two rate algorithms (BB and RMPC) and for video **D** with BB. Comparing **A**-BB to **A**-RMPC, we see that for **A**-BB, C3 achieves a higher QoE than C5 especially at the lower tail, while C5 is better for **A**-RMPC. Similarly, comparing **A**-BB and **D**-BB reveals that for the same BB rate algorithm, C3 achieves better tail performance than C5 for video **A**, while for video **D**, C5 is marginally superior.

Note that just encoding multiple different constant-length segmentations and making them available to clients to choose from adaptively, as suggested in past work [34], can address some of the above issues, but at huge expenses: if all of C1-

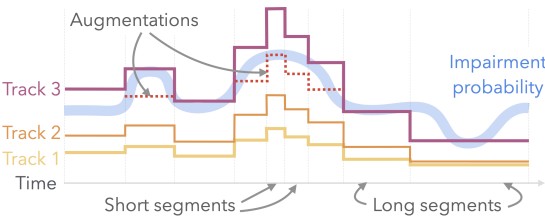

**Fig. 4:** *We can segment vulnerable parts of the video into shorter segments, and augment them with additional tracks.*

C5 were made available, the content provider's storage and caching expense would be $5\times$ larger.

In contrast to the above approaches, SEGUE consistently achieves higher QoE, as shown in Fig. 3, with only modest (under 10%) overhead in terms of additional bytes encoded.

## 4 SEGUE design
SEGUE tunes variable segment length and variable numbers of tracks across a video's segments to improve streaming quality. It does so in a manner that accounts for each segment's playback context and the rate adaptation algorithm. SEGUE uses the following inputs:

- A raw video whose encoding SEGUE will modify.

- The bitrate ladder the provider has designed for the video. This specifies the average bitrates of the different tracks.

- A target QoE function to optimize for.

- The rate adaptation algorithm the provider uses.

Given these inputs, SEGUE *segments* the video into variable-length segments, and *augments* some segments with added bitrate tracks. Fig. 4 shows a schematic of SEGUE's outputs.

### 4.1 Segmentation
Segmentation of a video must produce variable-length segments that should improve client QoE for the given ABR. SEGUE must output a segment sequence for every bitrate track specified in the input bitrate ladder. Further, the segments must obey the constraints **C1** and **C2** from §2.1.

**Problem formulation:** We first describe the problem ignoring the multi-track aspect. In this setting, segmentation involves first running a standard video encoder implementing the provider's codec of choice. We run the encoder on three inputs: the raw uncompressed video, the average bitrate to encode a track for, and a maximum gap between key frames.

The encoder outputs a compressed video track of (roughly) the input average bitrate. We use this compressed track's key frame positions as an input for segmentation.

Each pair of successive key frames contains between them a video fragment. Our task is to decide which video fragments to combine together into segments for streaming. As we scan the video track from left to right and encounter a key frame, should this key frame demarcate the start of a new segment, or should we merge the video fragment between this key frame and the next into our current segment?

For certain simple optimization criteria, *e.g.,* minimize the number of segments while limiting the maximum segment length, the problem of finding the optimal segmentation can be framed elegantly as a dynamic program. However, this is not the case for the more sophisticated optimization criteria SEGUE uses to improve QoE, as we discuss below. Thus, we use brute-force search over a limited horizon, $k$, of future key frames: we allow each binary decision for each key frame, *i.e.,* merge with the previous segment or not. Each of the $2^k$ outcomes is a candidate segment sequence. SEGUE then uses one of two broad methods for assigning value to each segmentation, and choosing the best.

**Intuitive heuristics:** Segments that are too long or too short, or have too many bytes or too few bytes are undesirable. For instance, segments with too many bytes will increase the likelihood of rebuffering while they are fetched. Similarly, segments that are too short in their playback time will cause too many requests to the video server, and incur larger transport and application-layer protocol overheads. Thus, to prevent our segmentation from producing such undesirable choices frequently, we can penalize it for such segments. We frame three heuristics that penalize deviations from target values, where the target is defined in terms of:

- *Time*: Segments that are too long or too short in terms of their playback time in seconds are penalized.

- *Bytes*: Segments that have too few or too many bytes are penalized. The target number of bytes must be set based on the track and the video.

- *Time + Bytes:* Segments that are too long or too short are penalized, but there is an additional penalty if a segment exceeds a byte threshold.

In each case, we evaluate the $2^k$ segmentations and pick the one that minimizes the penalty for deviating from its target in terms of time, bytes, or a combination, as noted above. Only the first segment of the chosen segmentation is final; the procedure continues from the first key frame after it, in a sliding window manner, until all key frames are processed.

To extend to multiple tracks, we run the above process for the highest-bitrate track. With the segment boundaries known, we encode all the other tracks by asking the encoder to impose key frames at the segment boundaries.

**Simulation-based assessment:** Instead of applying heuristics derived from our intuitions, we can also just evaluate each of our candidate segmentations for our target QoE function and ABR adaptation algorithm, across a set of diverse test traces, and pick the one that performs the best.

Besides the philosophical distinction from the intuitive heuristics approach, a simulation-based approach requires a change in methodology. We can no longer start with a single-track approach and later use the segment boundaries to inform segmentation of other tracks. Instead, we need all tracks to be segmented simultaneously in progression, because the simulation will involve switching between tracks.

To this end, we again have the encoder encode the highest-bitrate track in the same manner as before. However, instead of optimizing segmentation using only this track, we also ask the encoder to encode all the other tracks enforcing all the key frames to be the same as those in the highest-bitrate track.[1] We thus have all the tracks available simultaneously to perform segmentation on using a simulation.

For any candidate sequence of segments, $\mathbb{S}$, out of the $2^k$ options, the simulation, *Sim*, runs as follows:

1. For a set of network traces, we run the ABR on $\mathbb{S}$.[2]

2. For each trace, we compute the QoE, considering $\mathbb{S}$'s segments and all segments fetched preceding $\mathbb{S}$.

3. Across traces, QoE is aggregated based on a desired function, *e.g.,* the mean or an arbitrary specified percentile.

Across candidate sequences, the one that achieves the highest aggregated QoE across traces is selected, and its *first* decision — merge or not — is used. A merge decision results in the video fragment being added to the previous segment. If the decision is to not merge, the previous segment is closed. The simulation continues over the next $k$ key frames, in the same sliding window manner as for the heuristics.

The above *Sim* approach is effective in many settings, but it can be myopic due to its limited lookahead, ignoring long-term effects of a segmentation strategy. However, a longer lookahead, $k$, is computationally expensive. We thus also test a *WideEye* strategy that combines *Sim* with the preceding intuitive heuristics: we use a longer lookahead, but we: (a) filter out candidate sequences using the *Time + Bytes* heuristic; and (b) slide the window forward by multiple keyframes, thus freezing multiple decisions in each step instead of just one decision.

### 4.2 Augmentation

Augmentation aims to identify parts of a video vulnerable to streaming impairments, and add more bitrate tracks for their segments at appropriate bitrates.

Augmentation treats the video tracks and segmentation as inputs. The input bitrate ladder comprises a set $\mathbb{B}$ of tracks.

---

[1]The impact of this imposition of keyframes on encoding efficiency compared to a standard GOP method is small: for our settings, the VMAF loss and the bytes overhead are negligible, both changing by under 0.03%.

[2]Some algorithms, like Robust MPC [37], plan their decisions by looking ahead at several future segments. If this lookahead goes beyond the segments in $\mathbb{S}$, our simulation uses the future video *fragments* for this lookahead.

The segment set $\mathbb{S}$ can be comprised of segments as today, equal-length, or be the output of our above segmentation. A video $\mathbb{V}$ can be concisely described as a set of $\mathbb{B} \times \mathbb{S}$ segments across tracks. We define an augmentation technique, $\lambda$, as a function $\lambda : \mathbb{V} \rightarrow \mathbb{A}$, with $\mathbb{A}$ being the set of new tracks added, with each element $a \in \mathbb{A}$ describing the position of $a$ in the video and the average bitrate of $a$. We describe four augmentation functions, $\lambda_v$, $\lambda_b$, $\lambda_{bv}$ and $\sigma_{bv}$, which use different heuristics to identify segments to augment.

$\lambda_v$ **based on VMAF drops:** Despite the freedom afforded by VBR coding, complex scenes can end up with lower bitrate than needed to maintain perceptual quality. Our $\lambda_v$ heuristic attempts to augment such parts of a video. Consider the $i$th video segment on the $j$th bitrate track, $s_{i,j}$. If the VMAF of $s_{i,j}$ falls below the median VMAF across segments in track $j$ by a tolerance threshold, then $s_{i,j}$ is marked for augmentation. We add an additional encoding for the $i$th video segment, using the average of the bitrates of $s_{i,j}$ and $s_{i,j+1}$.

$\lambda_b$ **based on bitrate peaks:** Recall from Fig. 2(c) that segments corresponding to complex video scenes can be encoded at much higher bitrate than the average, with large gaps between the bitrates of successive tracks. This can make streaming these segments difficult, often requiring ABR adaptation to either switch to a lower track, or increase the risk of rebuffering.[3] By augmenting such segments with additional bitrates, we can offer greater flexibility in ABR adaptation.

Consider the $i$th segment on the $j$th track, $s_{i,j}$. If the bitrate of $s_{i,j}$ is above the average for track $j$ by more than a tolerance threshold, $s_{i,j}$ is marked for augmentation. We then add an additional encoding for the $i$th segment, with the bitrate corresponding to the average for track $j$. The VMAF of this newly added segment will lie between that of $s_{i,j-1}$ and $s_{i,j}$.

$\lambda_{bv}$ **using both bitrate and VMAF:** Not all segments chosen by $\lambda_b$ are challenging to stream in the same way. For instance, $s_{i,j}$ may have a high bitrate compared to track $j$'s average, and would be augmented by $\lambda_b$. However, if the VMAF loss from downloading $s_{i,j-1}$ instead of $s_{i,j}$ is relatively small then $s_{i,j}$ does not necessarily need augmentation. This is what our $\lambda_{bv}$ heuristic does: $s_{i,j}$ is only augmented *if* its bitrate is large relative to track $j$ *and* there is a substantial difference in the VMAF of $s_{i,j}$ and $s_{i,j-1}$.

$\sigma_{bv}$**, based on simulation:** Like for segmentation, we design an augmentation approach based on simulation. Unfortunately, the search space for augmentation is even larger than segmentation: each segment in our lookahead horizon can be augmented between every pair of its successive bitrate tracks. With just 6 bitrate tracks and a lookahead of 5 segments, the search space expands to $\sim$1 billion iterations per simulation step. We limit this scope substantially by using the $\lambda_{bv}$ heuristic as the basis: at each simulation step, we

limit augmentation candidates to ones suggested by $\lambda_{bv}$. Each parameter configuration of $\lambda_{bv}$ (in terms of the VMAF difference and bitrate difference thresholds) yields one candidate augmented segment sequence. We simulate ahead with each candidate sequence, as well as with the unaugmented (default) sequence. For each candidate, we quantify its QoE improvement compared to the default normalized by the overhead in terms of bytes added by that augmentation sequence. We pick the top scoring candidate, and continue this process from the next segment.

## 5 Implementation

We implement both the offline video chunking and online rate adaptation components to evaluate SEGUE.

### 5.1 Offline video chunking

SEGUE's chunking pipeline is implemented in Python3, and makes use of *ffmpeg* and libraries for standard codecs. Note that we are not devising new codecs, compression algorithms, or video formats; instead it is our explicit goal to stick to current, widely used codecs, as their implementations are heavily optimized, with mature provider-side pipelines, and client-side decoding often offloaded to hardware. Rather, we use the same codecs in a manner different from that in ABR video streaming today, as described in §4. Since the availability of raw video data sets of sufficient length for interesting ABR adaptation is limited, we instead use 4K compressed video as a stand in for raw video, and then limit our work to resolutions 1440p and lower. The bitrate ladders we adopt throughout are from Bitmovin [1], but arbitrary other bitrate ladders, including those customized per title [3] could be used as input. We also follow the guidance in that reference for encoding, using the recommended maximum bitrate of a track, *i.e.,* $1.75\times$ its average. Throughout, we use two-pass encoding, as is typical in the industry [25].

**Segmentation:** We implement the constant-length segmentation strategy common today as the baseline. We use *ffmpeg-libx264*, which allows us to specify certain key frame locations precisely, with the encoder potentially inserting additional key frames as necessary. This enables straightforward implementation of both the constant-length segmentation, as well as our segmentation heuristics (§4.1). We use the following configuration parameters:

- The *constant*-length baseline uses 5s segments.

- The lookahead of video fragments for all our segmentation methods except *WideEye* is $k = 5$, such that each iteration evaluates all $2^5$ segmentations of these fragments.

- For *Time*, the target segment length is 5s, with a penalty of 20% per second for deviations.

- For *Byte*, the bytes-per-segment target is the average bytes in 5s of video; excess bytes incur 20% penalty.

- For *Time+Bytes*, besides the time penalty, the byte penalty is also imposed for segments with too many bytes.

---

[3]CAVA [26], which is designed to carefully account for variable bitrate, also experiences this trade-off, but it is better at navigating it than non-VBR-optimized algorithms (§8). Our goal is to improve the trade-off itself.

- For *Sim*, the QoE of a candidate sequence is aggregated as the mean QoE across traces.

- *WideEye* has a lookahead of 10 keyframes instead of 5, and a decision window of 5 instead of 1. We only simulate the 32 best candidate sequences as ranked by *Time+Bytes*.

The penalties thresholds have been tuned to better work with our encoding settings, while the simulation lookaheads have been picked to keep reasonable computational time.

**Augmentation:** Our augmentation strategies are simple to implement as described in §4.2 using *ffmpeg-libx264*: regardless of the particular augmentation function, we merely need to encode a specific time range of video at a particular average bitrate, as a standalone segment. The different augmentation strategies are configured as follows:

- $\lambda_v$ : segments are augmented when their average VMAF is $\geq 8$ points lower than the median for their track (a value that corresponds to a bump from 720p to 1080p on a 4k TV [21]).

- $\lambda_b$ : segments are augmented when their bitrate is $\geq 10\%$ above than the average bitrate for their track. This leads to an aggressive augmentation strategy, intended to provide an upper bound QoE gain of this general method.

- $\lambda_{bv}$ : We tested $\lambda_{bv}$ for several different configurations. Segments are augmented when their bitrate is $\geq B\%$ above the average for their track *and* the VMAF difference between their track and the one below exceeds $V$ points, being V in $\{5, 6, 7, \ldots, 14\}$ and B in $\{5\%, 10\%, 15\%\}$.

- $\sigma_{bv}$ : We generate 30 candidate sequences by running $\lambda_{bv}$ with these thresholds — VMAF difference in $\{5, 6, 7, \ldots, 14\}$ and bitrate difference in $\{5\%, 10\%, 15\%\}$.

For augmentation, as well as for later evaluation, we need to compute the perceptual quality score, VMAF, for a video segment. We use the code made available by Netflix [13, 20]. For computing our augmentation strategies, we use the VMAF 4K model, while for our evaluation, we additionally evaluate the VMAF HDTV and VMAF Mobile models.

## 5.2 Online playback and rate adaptation

The approach we explore deliberately steps outside the DASH standard [31], with constant-length segments and a fixed number of bitrate tracks per segment. We thus modified the DASH player to support SEGUE. However, we use this implementation only to verify the fidelity of an orders-of-magnitude faster simulator, which we use for extensive experimentation. The simulator is implemented following the methodology described by the Pensieve authors [18]. Appendix D details the DASH player implementation, demonstrating that it achieves results near-identical to the simulator.

We simulate both the network and the player state. The network environment takes as an input a trace of bandwidth over time, and simulates the download of segments. The link RTT is set to 80ms in our experiments. The player simulator interacts with the network environment by requesting the download of a certain video segment from a certain track (as decided by the adaptation algorithm), and adds the segment's playback duration to the playback buffer. Meanwhile, it also drains the buffer. The maximum playback buffer size is limited to 60s; if the buffer is full, the player waits before downloading additional video segments. The number of seconds of buffered video needed before the player starts playback is set to 10s, following prior work [26].

The simulator logs rebuffering time and the downloaded tracks, allowing us to calculate QoE metrics in hindsight.

We evaluate SEGUE on four different ABR algorithms:

**Rate-based adaptation (RB)** tries to fetch the next segment at a bitrate matching the estimated network bandwidth. We adapt the simple, demo implementation of this strategy provided by Bitmovin [5]. This approach requires no modification to use SEGUE's modified encoding.

**Buffer-based adaptation (BB)** makes decisions entirely based on the player buffer state [8]. Briefly, BB uses two parameters: reservoir, $r$, and cushion, $c$. If the buffer size, $b$, is smaller than $r$, the lowest-bitrate track is used. If $b \geq r + c$, the highest bitrate is used. For $b \in [r, r+c]$, bitrate tracks are (roughly) linearly matched to the buffer sizes.

BB requires modest changes with SEGUE. Besides changing $r$ dynamically to adapt to variable bitrate coding as suggested in the original paper [8], we also bound $r$ by a minimum of 8 seconds to account for variable-length segments. Further, when $b \in [r, r+c]$, we first map $b$ to a bitrate range based on the unaugmented bitrate tracks available, but if additional tracks were made available by SEGUE, we further linearly match within this range to the appropriate track. These minor implementation tweaks substantially improve performance compared to a naive implementation.

**Robust MPC (RMPC)** uses control theory [37]. It uses the bandwidth estimate, current buffer size, and features of upcoming segments, to plan a sequence of requests based on the expected reward. It is flexible enough to incorporate knowledge about varying segment lengths and augmented bitrates. We tested two versions of RMPC: (a) RMPC-oblivious, where, as in [26], we modified the RMPC reward function to work with the instantaneous segments bitrates (rather than fixed weights); and (b) RMPC-aware, where we modified the reward function to account for VMAF score rather than bitrate. For both versions, to accommodate segments of different length, we weigh each segment's bitrate gain by its length.

## 6 Evaluation methodology

We evaluate our approach across network traces used in past work on ABR streaming, and test several videos.

**Network traces and VMAF:** We use a mix of ~600 traces across broadband 4G, HSDPA, and 3G networks [4, 29, 32]. The mean throughput of these traces spans from 350 Kbps to 60 Mbps. We divide the evaluation traces into three buckets:

| ID | Duration [mm:ss] | FPS | Content type |
|----|------------------|-----|--------------|
| A | 3:21 | 24 | 3D cinematic |
| B | 3:25 | 25 | Music video |
| C | 6:34 | 25 | Comedy |
| D | 3:51 | 25 | Festival |
| E | 4:42 | 30 | Action movie |
| F | 5:49 | 30 | Cooking tutorial |
| G | 2:33 | 30 | Sports (long-take-shot) |
| H | 2:40 | 30 | Sports (highlights) |
| I | 2:39 | 24 | Underwater |
| L | 3:16 | 30 | Drone footage |
| M | 2:40 | 30 | Video game |

**Table 1:** *An overview of our video dataset. Videos **C** and **D** lie at the extremes of highly stable and unstable in terms of perceptual quality over time within one track.*

- SLOW, containing traces with mean throughput <1.5 Mbps.

- MEDIUM, with mean throuput between 1.5 and 4 Mbps.

- FAST, with mean throuput >4 Mbps.

**Train and test separation:** Our simulation-based methods are data-driven. We use only 20% of the above ~600 traces to make segmentation and augmentation decisions.

For robustness, we test not only on the unseen 80% of traces from the above set, but also on an entirely different trace distribution from the Puffer project [33]. We sampled Puffer traces from Dec. 2020 to May 2021, arbitrarily using data from the 5$^{th}$ of each month. We retain only those traces that are longer than 2 minutes, corresponding to 64% of Puffer traces. 3.1% of these traces fit the SLOW class, 5.6% MEDIUM and 91.3% FAST. Our test set uses an equal number of Puffer and non-Puffer traces, *e.g.,* half of the SLOW test set is a random sample of SLOW-class Puffer traces, while the other half is from the 80%-unseen data from the other trace distributions mentioned above. Note again that this implies that none of the test data has been used in decision-making, and that half of it comes from an entirely different data source. (Limiting our evaluation to only the Puffer trace dataset only makes the results more favorable to SEGUE.)

Unless noted otherwise, we evaluate SLOW traces on the VMAF mobile model, MEDIUM on HDTV, and FAST on 4K.

**Videos:** We use a set of 11 videos with different content, downloaded from YouTube, listed in Table 1. These videos are available in 4K, which we use as "raw" (§5.1), and then run experiments for 240p, 360p, 480p, 720p, 1080p and 1440p.

**QoE function:** Unfortunately, with variable-length segments, we cannot use the QoE function used in past work as is, because it aggregates QoE metrics across *equal-length* segments (§2.1). Instead, we adapt the formulation to sum QoE per unit time, at a granularity of 1s. This is small enough to capture any impact from our use of smaller segments.

This implies that we have to adjust the weights λ, β, and γ

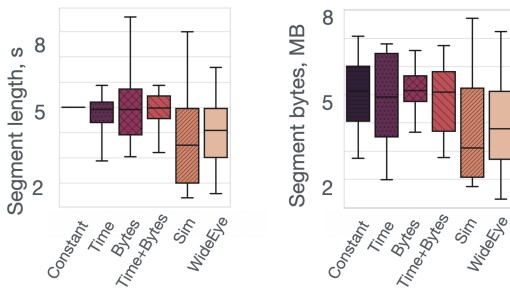

**Fig. 5:** *Segment characteristics for different schemes — video **A**, RB. Boxes show mean and quartiles, whiskers are 5/95-percentile.*

for the QoE components corresponding to VMAF, rebuffering, and VMAF switches respectively: the original weights used in past work, are for 4 second segments, and using that same formulation on 1s intervals would effectively assign 4× the importance to VMAF. We thus scale λ by $\frac{1}{4}$. Further, as we compare schemes with different segment lengths, we cannot ignore the startup phase: doing so would benefit schemes that fetch longer segments, as they would build up more buffer. We simply account for the initial phase in the same manner as any other segment, incurring a rebuffering penalty until the first segment is downloaded and played.

To use VMAF instead of bitrate as in the Robust MPC work [37], we also need to adapt the weight for rebuffering. MPC's QoE function, drawn from measurement work, penalizes each second of rebuffering, *i.e.,* β, as equivalent to losing 4s of full-resolution bitrate. For VMAF, full-resolution translates to a value of 100. Thus we use β = 100.

We decrease the switching penalty, γ, from 2.5 to 1. With a 1 sec cadence for QoE evaluation, we measure switching more often than prior work. This accounts for intra-segment changes in VMAF, and penalizes any additional switching caused by our potentially shorter segments. (Using prior work's γ = 2.5 setting only improves SEGUE's results.)

In any case, SEGUE can be run with arbitrary QoE functions.

## 7 Results

We first describe the improvements from SEGUE for video **A** and RB adaptation. This helps draw out intuition in detail. Later, we evaluate SEGUE across 11 videos, 4 adaptation algorithms, hundreds of network traces, and 3 VMAF models. We then compare SEGUE's performance with the closest related works, and discuss its computational cost.

### 7.1 SEGUE's segmentation

Fig. 5 shows the characteristics of the segments produced by different approaches. *Time* and *Bytes*, by design, produce segments of similar duration and bytes respectively to *Constant*. However, by constraining only one factor, they introduce large variations in the other. *Time+Bytes* constrains both, and is thus conservative in its segmentation. *Sim*, with no direct constraints, naturally produces segments with the greatest variability. Consider, *e.g.,* a low-complexity credits scene, for which *Sim* may produce a very long segment to pre-

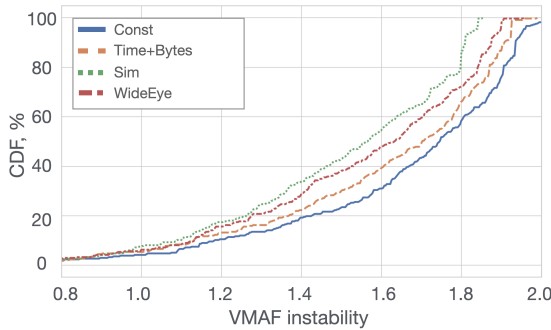

**Fig. 6:** *VMAF fluctuations: video **A**, RB adaptation, SLOW traces.*

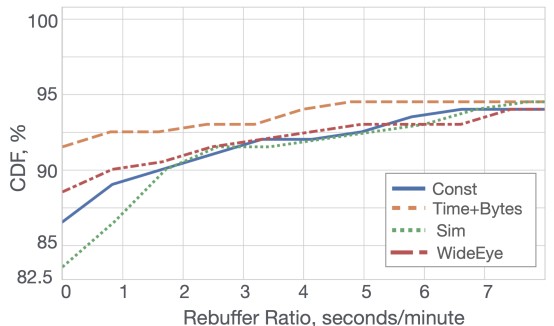

**Fig. 7:** *Rebuffer ratio: video **A**, RB adaptation, SLOW traces.*

vent RB from incurring switching penalties in QoE. *WideEye* strikes a balance, allowing greater freedom in segmentation than *Time+Bytes*, but trimming out *Sim*'s extreme, myopic choices. *Time* and *Bytes* are the least performant schemes, with obvious reasons, so we omit further discussion of these.

We measure VMAF fluctuation as the average change in VMAF per second of playback. We normalize this by the average VMAF fluctuation experienced by *Constant* across our full cross product of videos, traces, and rate adaptation algorithms. We calculate rebuffer ratio as the sum of seconds of rebuffering experienced during playback divided by video duration, and reported in seconds per minute (s/m).

Fig. 6 shows VMAF fluctuations across the SLOW traces. *Sim* achieves the most stable streaming, at the cost of higher rebuffer ratio (by 1 s/m) compared to *Constant*. *Time + Bytes* improves VMAF stability modestly, while simultaneously improving rebuffer ratio by 0.7 s/m compared to *Constant*.

*Sim*'s numerous overly long segments, which help drive RB away from the frequent track switching it is prone to, result in an increased risk of rebuffering (Fig. 7). This is a consequence of its short-term, myopic decision making, which does not account for future risk of rebuffering.

*WideEye* strikes the more favorable tradeoff here, not only improving stability substantially, but also limiting rebuffering to only 0.1 s/m higher than *Constant*, compared to *Sim*'s 1 s/m.

Changes in delivered VMAF are modest, with *WideEye* improving over *Constant* by ~0.8% for SLOW/MEDIUM traces.

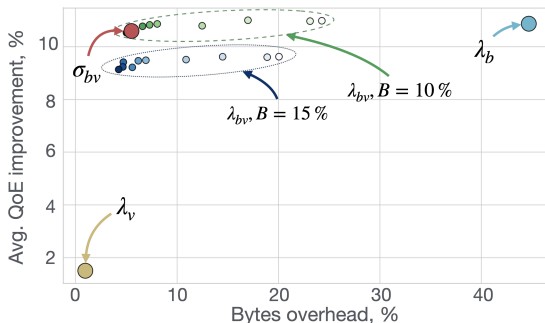

**Fig. 8:** $\sigma_{bv}$ *improves QoE with only small byte overheads, as does* $\lambda_{bv}$ *with appropriate parameters.* $\lambda_{bv}$ *is shown with* $V \in \{5, 6, 7, \dots, 14\}$ *for both* $B = 15\%$ *and* $B = 10\%$. *(B = 5% is similar to B = 10%.)*

**Takeaway:** Intuitive heuristics like *Time+Bytes* conservatively perform segmentation, avoiding risks like overly long or large segments. On the other hand, a short-term simulation approach can be overly aggressive, and increase longer-term rebuffering risk. Merging intuition with a longer-term simulation horizon strikes a favorable tradeoff.

### 7.2   SEGUE's augmentation

We next examine QoE improvements for video **A** with RB, by comparing *Constant* to SEGUE with *WideEye* segmentation coupled with each of our 4 augmentation heuristics in Fig. 8.

$\lambda_v$ (bottom-left, yellow) and $\lambda_b$ (top-right, cyan) show extreme points: the former augments too few segments and results in negligible improvements, while the latter augments too many segments (incurring more than additional bytes for encoding) to achieve its substantial QoE gains.

Our simulation-based strategy, $\sigma_{bv}$, (top-left, large red dot) achieves both high QoE improvement and low overhead in terms of bytes, due to its careful choices of which segments to augment. Mean QoE improvements compared to *Constant* are 24.9%, 4.1% and 1.4% for SLOW, MEDIUM, and FAST traces respectively, at the cost of 5.5% of more bytes encoded.

We also find that for our QoE reward and byte overhead definitions, $\lambda_{bv}$ achieves similar results as $\sigma_{bv}$, if $\lambda_{bv}$'s parameters are appropriately tuned (specifically, using a bitrate threshold, $B = 10\%$, and a VMAF threshold, $V = 13$ or $14$).

We find that a small additional amount of bytes encoded for augmentation improve VMAF stability and reduces rebuffering, together with modest improvements on the average VMAF delivered. It is worth noting that augmentation is not as simple as "augment more bytes for higher QoE"; in fact, there are several heuristics that incur higher overhead, with lower QoE benefit, *e.g.*, compare several of the $\lambda_{bv}$, $B = 15\%$ results in Fig. 8 to $\sigma_{bv}$.

**Takeaway:** The simulation approach, by explicitly trading off QoE improvements with their cost, appreciably improves QoE at low overhead in terms of additional encoding and storage. At the same time, a careful tuning of parameters for a static policy can produce comparable results.

| ABR | SLOW | | MEDIUM | | FAST | |
|---|---|---|---|---|---|---|
| | Mean | 95% | Mean | 95% | Mean | 95% |
| BB | 5.2% ↑ | 12.2% ↑ | 5.3% ↑ | 5.7% ↑ | 4.7% ↑ | 10.3% ↑ |
| RB | 5.6% ↑ | 18.4% ↑ | 7% ↑ | 8.3% ↑ | 4.5% ↑ | 7.8% ↑ |
| RMPC-O | 1.2% ↑ | -4.6% ↓ | 1% ↑ | -1.1% ↓ | -1.3% ↓ | 7.4% ↑ |
| RMPC-A | 4% ↑ | 9.6% ↑ | 3.2% ↑ | 17% ↑ | -1.5% ↓ | -0.5% ↓ |

**Fig. 9:** *VMAF stability improvements divided by trace set and ABR. As expected, improvements are more significant for RB, given that no stability policy is implemented in the ABR.*

> **Takeaway:** More augmentation does not always improve QoE; rather segments to be augmented need careful choice.

## 7.3 The impact of the adaptation algorithm

**Segmentation:** Across our experiments, the largest improvements from segmentation come from VMAF stability during playback, typically with some improvements in rebuffering, and negligible changes in VMAF. However, the details differ across rate adaptation algorithms as we discuss next.

Fig. 9 shows the improvement in VMAF stability. For each adaptation algorithm, we compute the VMAF switching penalty term of the QoE aggregated across the cross-product of videos and traces; we then show the mean and 95th-percentile in the table cells. The largest improvements are seen for RB, followed by BB, and the two versions of R-MPC. For RMPC-oblivious we even see a deterioration, *i.e.,* higher VMAF switching by 4.6% at the 95th-percentile. It is worth noting that our segmentation simulations always use the VMAF 4K model to make decisions, while the evaluation uses different VMAF models for different trace buckets. If we evaluate using the VMAF 4K model, the result for RMPC-O is also positive, with 6.7% improvement. Using different VMAF models during segmentation tuning results in different weights for rebuffering, VMAF, and VMAF switching (*e.g.,* the mobile model is the most permissive for VMAF, weighting rebuffering more), so it is an open question as to how to optimize robustly against these differences.

For rebuffering, the differences from SEGUE's segmentation are smaller, with meaningful differences only at the tail. This is inherent to rebuffering: it is a corner case, as most rate adaptation approaches are conservative enough to avoid it in the typical case. For BB and RB, the number of traces for which rebuffers occur is cut by 1.3% for both, while for RMPC-O and -A, 0.2% and 1.3% more traces see rebuffers with SEGUE's segmentation compared to *Constant*.

We dissected the tail rebuffering and switching of SEGUE's segmentation with RMPC more deeply. RMPC plans for a limited lookahead of segments (5 in that paper), and when SEGUE produces several short segments, the lookahead becomes more and more myopic in terms of time, thus causing poor long-term planning. Thus, RMPC's implicit assumption that a certain number of segments comprises a long-enough future planning horizon is contradicted in SEGUE. Unfor-

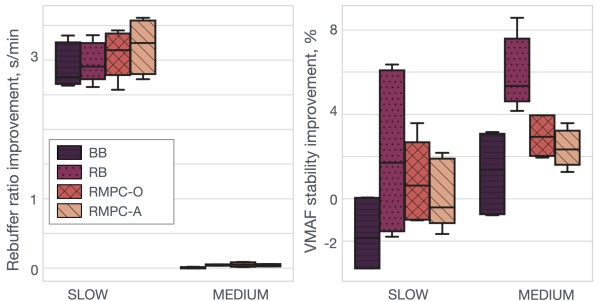

**Fig. 10:** *Improvements from adding $\sigma_{bv}$ to WideEye. We average metrics across traces, and show their distribution across videos. Boxes show mean and quartiles, whiskers are 5/95-percentile.*

tunately, increasing RMPC's lookahead is computationally expensive, so if the algorithm is not modified, there are two solutions: (a) disallowing series of short segments; and (b) instead of optimizing for the mean in SEGUE's segmentation, as we currently do, optimizing for higher percentiles (see §4.1). That *Time+Bytes* rebuffers on 1.9% and 2.1% fewer traces for RMPC-O and RMPC-A shows promise for strategy (a).

We also briefly illustrate the specificity of SEGUE's segmentation to different rate algorithms with experiments on video **A**: tuning segmentation using WideEye for RB and then using RMPC-O adaptation online actually *degrades* QoE by 7% compared to Constant, while correctly tuning segmentation for RMPC-O *improves* QoE by 6% compared to Constant.

> **Takeaway:** A mismatch in what rate algorithm segmentation is tuned for versus used with can degrade QoE.

> **Takeaway:** Segmentation's interactions with rate adaptation algorithms that implicitly or explicitly assume constant length segments require additional effort to either modify such rate algorithms, or SEGUE's interaction with them.

**Augmentation:** Augmentation typically improves all three QoE metrics, at the cost of a small provider-side compute and storage overhead. The gains are largest for rebuffering and switching, with smaller improvements for VMAF.

We compare *WideEye* with and without $\sigma_{bv}$ augmentation. For each of our 11 videos, we calculate the changes in the average metric across traces, *i.e.,* rebuffer ratio (in sec per min), VMAF stability (in percentage). The results shown in Fig. 13 are the distribution of these improvements from adding $\sigma_{bv}$ across the 11 videos. For the SLOW traces, rebuffering is reduced on average by >3 s/m for all algorithms. The improvements stem primarily from augmentation enabling finer-grained quality decisions especially at low-bitrate tracks. Even for RMPC, this compensates for the shorter lookahead. The differences are smaller for FAST traces (as expected), which are omitted in the plot.

Rebuffering improvements for RB are more limited because having more choices sometimes enables more aggressive behavior in RB, where the estimated rate has greater

chances of more closely matching an available bitrate. For the same reason, VMAF switches improve more for RB: it aggressively matches bitrate to rate estimates, and having more choices makes the fluctuations smaller.

VMAF gains are small in the aggregate, but this is a bit misleading: in many cases, augmentation improves VMAF noticeably (*e.g.,* ∼10%) for parts of playback, but these 'local' gains are suppressed in the aggregate, as VMAF does not change much for most segments. (Fig. 15 in Appendix B highlights the locality of these improvements.) It is unclear to us how or if QoE functions should reward such local improvements.

The provider-side costs of $\sigma_{bv}$ augmentation are small across all 4 rate adaptation algorithm, with roughly 8% overhead in bytes encoded on average across videos.

> **Takeaway:** Augmentation substantially improves rebuffering and VMAF stability, especially in low-bandwidth conditions, while incurring modest costs.

## 7.4 The impact of the video

How much a video's complexity varies over time greatly affects how much SEGUE benefits it. For instance, video **C** shows, on each of its tracks, very little variation in VMAF and bitrate. This lack of substantial temporal variation leaves little room for optimization beyond constant-length segments with fixed tracks. For video **C**, our segmentation's improvements in VMAF stability are smaller than on the rest of our data, and in some cases, there is even a degradation in performance (3.1% and 2.4% on average over SLOW traces with RMPC-A and -O respectively).

The other extreme is video **D**, with frequent changes across scenes. (Appendix A Fig. 14 visually contrasts videos **C** and **D**.) For video **D**, even for FAST traces, *WideEye* improves VMAF stability by 12% on average for RB and BB.

For 3 videos in our dataset (video **B**, video **G** and video **I**), SEGUE's segmentation hurts the performance for both versions of RMPC. For video **B** and video **G** we have a degradation in terms of delivered VMAF, with average perceptual quality degradation of 3%. For video **B**, this degradation also appears in VMAF stability. (Augmentation partially makes up for this deterioration.) This is due to the behavior in the startup phase: for some videos, producing small segments at the beginning greatly slows down the ramp-up of RMPC to higher bitrates. Modifying the objective function in RMPC to account for startup would likely ameliorate this issue.

For Video **I**, however, SEGUE with RMPC-A substantially degrades performance, with a perceptual quality loss in FAST traces of 12% and a loss in VMAF stability of 23%. This behavior is caused by a quirk of the bandwidth estimation approach (which we left untouched from prior work [18]), whereby the RTT is also incorporated to the calculation of the bandwidth estimate. The impairment occurs when, at the beginning of a video, there are one or more segments comprising as little as a few kilobytes of data, *e.g.,* a few seconds

| | VMAF [%] | VMAF Instability [%] | Rebuffer Ratio [s/min] |
|---|---|---|---|
| GOP-1 | 2.2% ↑ | -43.4% ↓ | -0.6 ↓ |
| GOP-2 | 0.4% ↑ | -30.2% ↓ | -0.3 ↓ |
| GOP-3 | -0.06% ↓ | -21.8% ↓ | -0.24 ↓ |
| GOP-4 | 0.1% ↑ | -19.4% ↓ | -0.19 ↓ |
| GOP-5 | -0.02% ↓ | -14.2% ↓ | -0.20 ↓ |

**Fig. 11:** *Performance improvements of* SEGUE*'s segmentation strategy WideEye over the delivery of single GOPs, varying the GOP size.*

of a completely black screen or title screen. In this case, the bandwidth-dependent download time can be smaller than the RTT. Incorporating the RTT in the bandwidth estimation thus substantially underestimates bandwidth, and slows down the ramp up of video quality. A constant-length segmentation is immune to this bug, while *any* segmentation that allows smaller segments is affected by it. Simply separating the RTT estimate from the download time would eliminate this issue.

> **Takeaway:** SEGUE's benefits are larger for more complex video content. This could be used to build a meta-heuristic to decide whether or not to use SEGUE for a particular video.

## 7.5 Performance comparison with related works

We now compare SEGUE against the three closest related works. First, we compare SEGUE's *WideEye* segmentation strategy to [30], in which videos are segmented and delivered following GOP boundaries. Then, we compare SEGUE's $\sigma_{bv}$ augmentation strategy to CBF [27] and SIVQ [28]. Both solutions aim to remove *redundant* segments from the video representation. While CBF removes segments in order to be as close as possible to a target quality, SIVQ focuses on keeping the ones that differs enough in terms of perceptual quality.

### 7.5.1 Comparison with GOP delivery

In order to compare SEGUE's segmentation strategy to [30], we encoded our full dataset of videos varying the maximum GOP size from 1s to 5s, with a step of 1s. We then tested the streaming performance of transmitting each GOP separately against SEGUE's *WideEye* segmentation strategy across the cross product of network traces and ABRs. Results are summarised in Fig. 11.

Improvements in perceptual quality are significant only in the case of GOP-1, where SEGUE delivers in average 2.2% better quality.

SEGUE consistently reduces perceptual quality fluctuations. This is an intrinsic property of SEGUE: by deciding which GOPs to pack together (or not) depending on the video flow and ABR behavior, SEGUE is able to successfully stabilize the video stream compared to a fully fragmented solution. These improvements are major against a GOP length of 1s,

where SEGUE reduces instabilities by 43.4%, and become smaller, but still significant, when increasing the GOP size. VMAF instability reduction over the GOP-5 segmentation (that is, indeed, the GOP size in which SEGUE's *WideEye* is performed) is in average 14.2%.

SEGUE also substantially improves rebuffering ratio. Shorter segments, in fact, do not necessarily reduce the likelihood of rebuffering events, as they might mislead the ABR into poor buffer planning and, in general, greedier choices. Again, by tracking the video flow and ABR choices, SEGUE is able to reduce substantially the average rebuffering ratio, by 0.6 s/m in the case of GOP-1, and by around 0.2 s/m in the case of GOP-5.

Compared to GOP-5, SEGUE segmentation strategy's improvements in the linear QoE model utilised throughout this work are, in average, 4.2%. In the case of GOP-1, these improvements increase to 14%.

### 7.5.2 Comparison with CBF and SIVQ

We compare SEGUE's augmentation strategy $\sigma_{bv}$ to CBF [27] and SIVQ [28]. All the approaches are applied to SEGUE's *WideEye* segmentation. We offer to CBF and SIVQ **all** the available options, in other words the ones that SEGUE uses as standard (and non removable) and the augmented ones. We test CBF under three VMAF thresholds (40, 60, 80). We also modified the SIVQ algorithm to work with VMAF rather than PSNR, and we test it for three different thresholds (5, 10, 15).

In Fig. 12 we show the comparison with SEGUE's $\sigma_{bv}$ aggregated across our cross product of videos, ABRs and traces. Both SIVQ and CBF substantially reduce rebuffering and storage compared to SEGUE. This is expected, as both approaches (CBF more aggressively than SIVQ) almost entirely remove the 1440p track, and substantially cut down the 1080p track. This forces all the ABRs to perform safer choices, as the highest quality options are not available.

The removal of higher quality tracks has the drawback of reducing the delivered quality, in particular for FAST traces. This is an expected behaviour of both CBF and SIVQ, as they optimize for bandwidth and storage savings. The severity of this reduction depends on the selected threshold, with SIVQ-5 being the least affected: 0.4% degradation in average, with 1.5% degradation in FAST traces and 4.3% in the 95-th percentile of the best traces.

Compared to SIVQ-5, SEGUE improves the VMAF stability in FAST traces by 4.8%, an improvement that is coherent with the one experienced by introducing augmentation. For SLOW and MEDIUM traces, improvements in stability are not substantial. In any case, as explained in section §6, despite our trace set being balanced, 91% of traces in the analyzed Puffer set are classified as FAST. SEGUE's $\sigma_{bv}$ improves substantially in challenging scenarios without affecting the user experience in the most common setting.

Compared to CBF-40 and SIVQ-15, SEGUE's improvements in average on FAST traces for the linear QoE formulation utilised in this work are of 45% and 5.3% respectively. These improvements are still substantial if compared to CBF-80 (5%), and become small if compared to SIVQ-5 (>1%), due to the low weight on VMAF instability in our linear QoE formulation. VMAF instability is indeed the optimisation metric that SEGUE improves the most.

Last but not the least, SEGUE can be combined with both CBF and SIVQ. In case of CBF, one could pre-filter both standard and non standard options for a specific quality setting, and then run the SEGUE's optimization. Similarly, for SIVQ we could just present to SEGUE optimizer the video segments that are sufficiently different in terms of perceptual quality performance. However, since adding CBF or SIVQ to SEGUE would lead to a worse outcome on FAST traces, we did not include such a combination in our evaluation.

### 7.6 SEGUE's computational cost

We benchmarked SEGUE's computational performance for video **B**, as it strikes a good tradeoff between bitrate variability and keyframes frequency. The benchmark runs on an AMD Ryzen 9 3900x 12-Core processor and Ubuntu 20.04.3 LTS. Results are presented in Fig. 13 as a fraction of the total computation time and the video length. SEGUE's performance highly depends on the ABR algorithm's efficiency. SEGUE's computation time using fast ABR algorithms like rate or buffer based is comparable to the VMAF computation time. Using slow ABRs, like both version of R-MPC, takes considerably more time due to the state space exploration (a problem that has been tackled by the authors in [37], and that lead to the formulation of the more lightweight version Fast-MPC).

Compared to the H.264 encoding time, SEGUE's segmentation takes 3.5x more time with the fastest ABR. Nevertheless, SEGUE's segmentation times are comparable to the computational time needed by more recent codecs, like VP9 and AV1, that take significantly more time compared to H.264 (5x and 10x respectively [2]), while SEGUE's approach and costs are independent on the codec of choice.

SEGUE's current release is not optimized for runtime and written in Python3 using the multiprocessing module. This module uses expensive process based parallelism. A reimplementation in an unmanaged language with better multithreading support (like C++ or Rust) would likely offer at least an order of magnitude improvement in compute times, as for example Numba [11] discusses. Also, given the substantial amount of time that is spent on VMAF calculations, SEGUE could be extended to either work with computationally cheaper quality metrics (like PSNR or SSIM), or approaches like the one in [10] could be used for VMAF rate distortion curves prediction.

### 7.7 Summary of results

We evaluated SEGUE across 4 adaptation algorithms, 11 videos, and 3 trace buckets. SEGUE typically maintains aver-

| | VMAF [%] | | VMAF Instability [%] | | RR [s/min] |
|---|---|---|---|---|---|
| | Mean | Fast Traces | Mean | Fast Traces | Mean |
| CBF-40 | 46.7% ↑ | 105.2% ↑ | -44.4% ↓ | -117.7% ↓ | 0.5 ↑ |
| CBF-60 | 17.1% ↑ | 44.8% ↑ | -26.1% ↓ | -85.3% ↓ | 0.34 ↑ |
| CBF-80 | 3.9% ↑ | 12.3% ↑ | -7.2% ↓ | -33.1% ↓ | 0.15 ↑ |
| SIVQ-5 | 0.4% ↑ | 1.5% ↑ | -0.7% ↓ | -4.8% ↓ | 0.1 ↑ |
| SIVQ-10 | 1.3% ↑ | 4.7% ↑ | -2.3% ↓ | -14.2% ↓ | 0.15 ↑ |
| SIVQ-15 | 2.3% ↑ | 8.3% ↑ | -4.6% ↓ | -24.2% ↓ | 0.18 ↑ |

**Fig. 12:** *Performance improvements and degradation of* SEGUE*'s augmentation strategy* $\sigma_{bv}$ *compared to CBF [27] and SIVQ [28]. Rebuffering ratio comparison for FAST traces is neglibile, and thus it has been omitted.*

| ABR | Encode time | | VMAF time | | WideEye | $\sigma_{bv}$ |
|---|---|---|---|---|---|---|
| | STD | AUG | STD | AUG | | |
| RB | 2.9 | 3.2 | 12.6 | 14.9 | 10.2 | 2.9 |
| BB | 2.9 | 2.6 | 12.6 | 14.9 | 10.2 | 2.9 |
| RMPC-A | 2.9 | 3.5 | 12.6 | 14.9 | 26.3 | 31.3 |
| RMPC-O | 2.9 | 3.5 | 12.6 | 14.9 | 26.3 | 25.4 |

**Fig. 13:** *Benchmarking of* SEGUE*'s performance for video **B** as a ratio between the computational time and the video length.* SEGUE*'s brute force exploration time is heavily affected by the ABR algorithm efficiency.*

age VMAF, while reducing switching and tail rebuffering, at the expense of reduced VMAF for a small fraction of chunks. This is a highly favorable tradeoff for the QoE function.

We calculate QoE improvements as: $100 \cdot \frac{Q_{\text{SEGUE}} - Q_{Constant}}{Q_{\max}}$, where $Q_{\max}$ is the maximum achievable QoE. Comparing $Q_{\text{SEGUE}}$ and $Q_{Constant}$ directly would only show larger numbers. Across our result matrix, SEGUE's mean QoE improvement is 8.6%, with 36.5% improvement in the $5^{th}$-percentile. When limited to SLOW traces, these numbers are 22.1% and 111% respectively. For interested readers, a full tabulation of results is in Appendix C Fig. 17.

For context on SEGUE's QoE improvements, we can compare them to those for algorithmic improvements in rate adaptation. Across our traces, QoE for *Constant* improves by less than 2% when using R-MPC instead of BB. (This is in line with experiments in the recent Puffer work [36], providing validation for our evaluation.) Our improvements are larger than what Facebook measured [17] in testing reinforcement learning adaptation, where under 6% improvements for traces with sub-500 Kbps bandwidth (as much $3\times$ slower than our SLOW set) are reported as "substantial" for Facebook.

## 8 Discussion and Future Work

With SEGUE, we have only begun exploring new opportunities that arise from accounting for the temporal variations in video content and their interactions with online adaptation.

**Algorithmic work:** Much like for rate adaptation, where new algorithms continue to be devised, we fully expect SEGUE to set off a new thread on how best to optimize chunking. A particularly promising opportunity for segmentation lies in doing chunking online, whereby the client could adaptively request video in terms of keyframe boundaries, instead of being restricted to a particular offline chunking scheme. This ap-

proach can adapt chunking to both video content and network variations, without needing real-time reencoding.

**Co-design of encoding and adaptation:** While most ABR work treats video as an uncontrolled input and focuses on adaptation, we take the opposite perspective, treating rate adaptation as a given, and exploring how to modify video chunking. This obviously raises the question of how closely we could integrate offline encoding and online adaptation.

As our results show, it is non-trivial to tweak algorithms like RMPC, which bake in today's typical constant-length segmentation in their design, to work well with SEGUE. Going further, how would SEGUE interact with an adaptation algorithm like CAVA [26], which explicitly tackles variable bitrate encoding.[4] Does either reduce the other's utility? Or does CAVA's non-myopic behavior benefit from SEGUE's offline preparation, resulting in even larger benefits?

Likewise, on the encoding side, does video for which bitrates are tuned per scene, like Netflix has started doing [22], reduce the benefit of SEGUE's augmentation? Does it increase the benefit of SEGUE's segmentation? How do the answers to these questions depend on the adaptation algorithms used?

In the context of co-designing adaptation logic with SEGUE, the most straightforward next step would be to modify SEGUE itself to output a set of representations, both in space and time, and to modify ABR logic to select (online) between these representations. We plan to investigate this path in future work.

**Deployment considerations:** SEGUE requires rethinking some aspects of video delivery: (1) As different segments have different numbers of tracks available, any user interface elements for manually selecting a track (disabling adaptation) need to hide that difference and make background decisions accordingly; (2) While we don't expect the potentially frequent and minor tweaks in a provider's adaptation algorithm to have large effects, large changes to adaptation will need to be compatible with the video library's chunking, although this is not very different from today — constant length segmentation is just one (implicit) choice.

---

[4]Unfortunately, our requests for CAVA's code were unsuccessful.

## 9 Conclusion

SEGUE is the first work to investigate offline video chunking in a manner that accounts for the interactions of online rate adaptation with the temporal variability in video complexity. Besides showing promising performance improvements, especially for challenging settings involving complex videos or low-bandwidth conditions, it calls for closer integration of offline and online phases. We discuss several exciting open questions, and release our code to enable their exploration [14].

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

# A    Appendix: Video internal variability

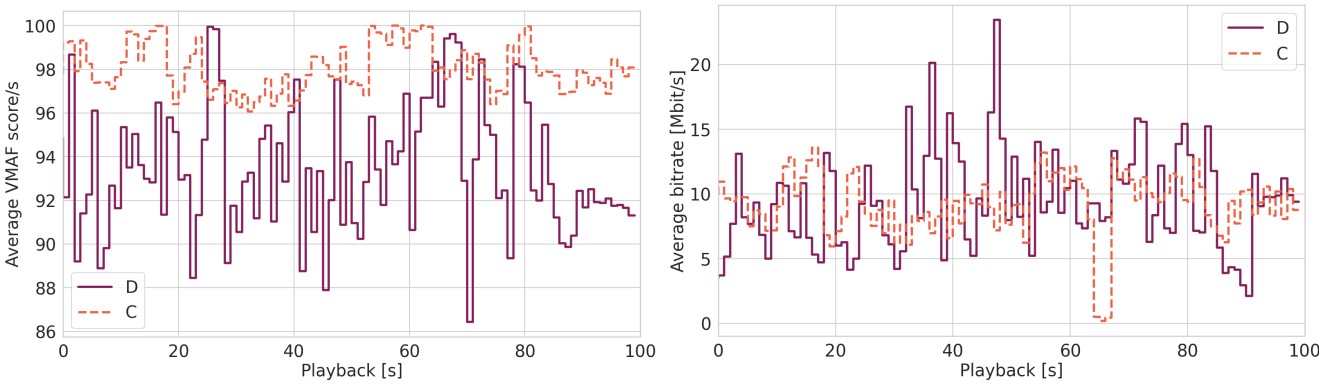

**Fig. 14:** *VMAF and bitrate comparison between video **C** and video **D** for the highest quality track of each. VMAF and bitrate are averaged per second, and shown for the first 100 seconds of playback. Video **C** exhibits much greater stability than video **D**.*

# B    Appendix: Locality of augmentation

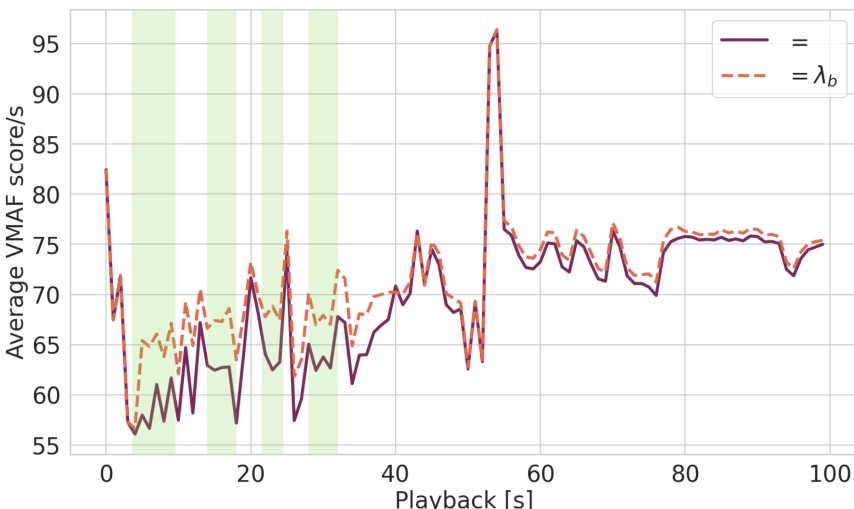

**Fig. 15:** *Augmentation yields local VMAF improvements: average VMAF/sec for video E using RMPC, Constant segmentation*

# C Appendix: Full tables of results

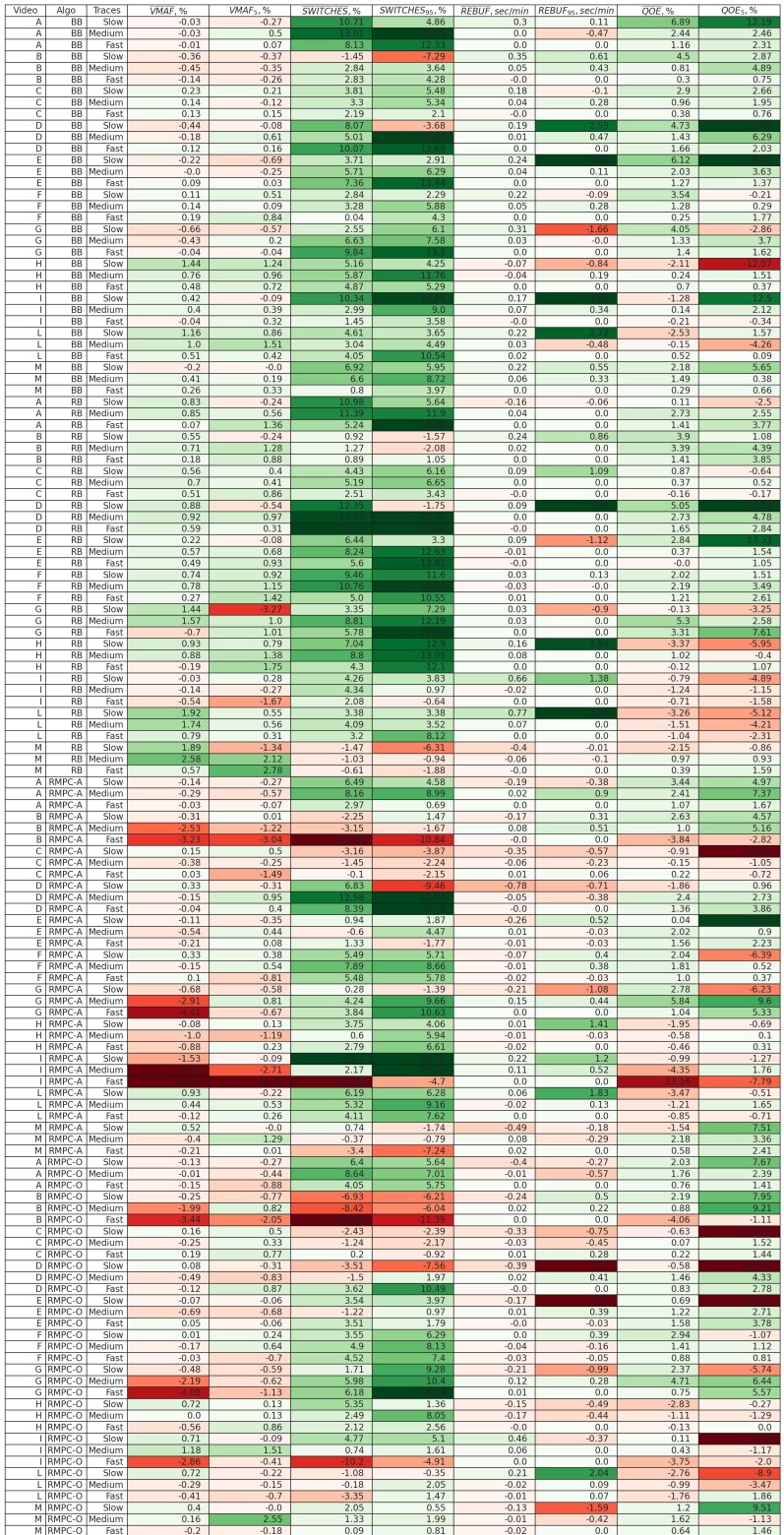

**Fig. 16:** SEGUE*'s results with WideEye segmentation and no augmentation, compared to Constant. This is a visual summary across 11 videos, 4 algorithms, and 3 trace sets, totaling to ∼220 days of streaming time. Each row is one combination of (video, algorithm, trace-bucket). Green colors are improvements, red being deterioration.*

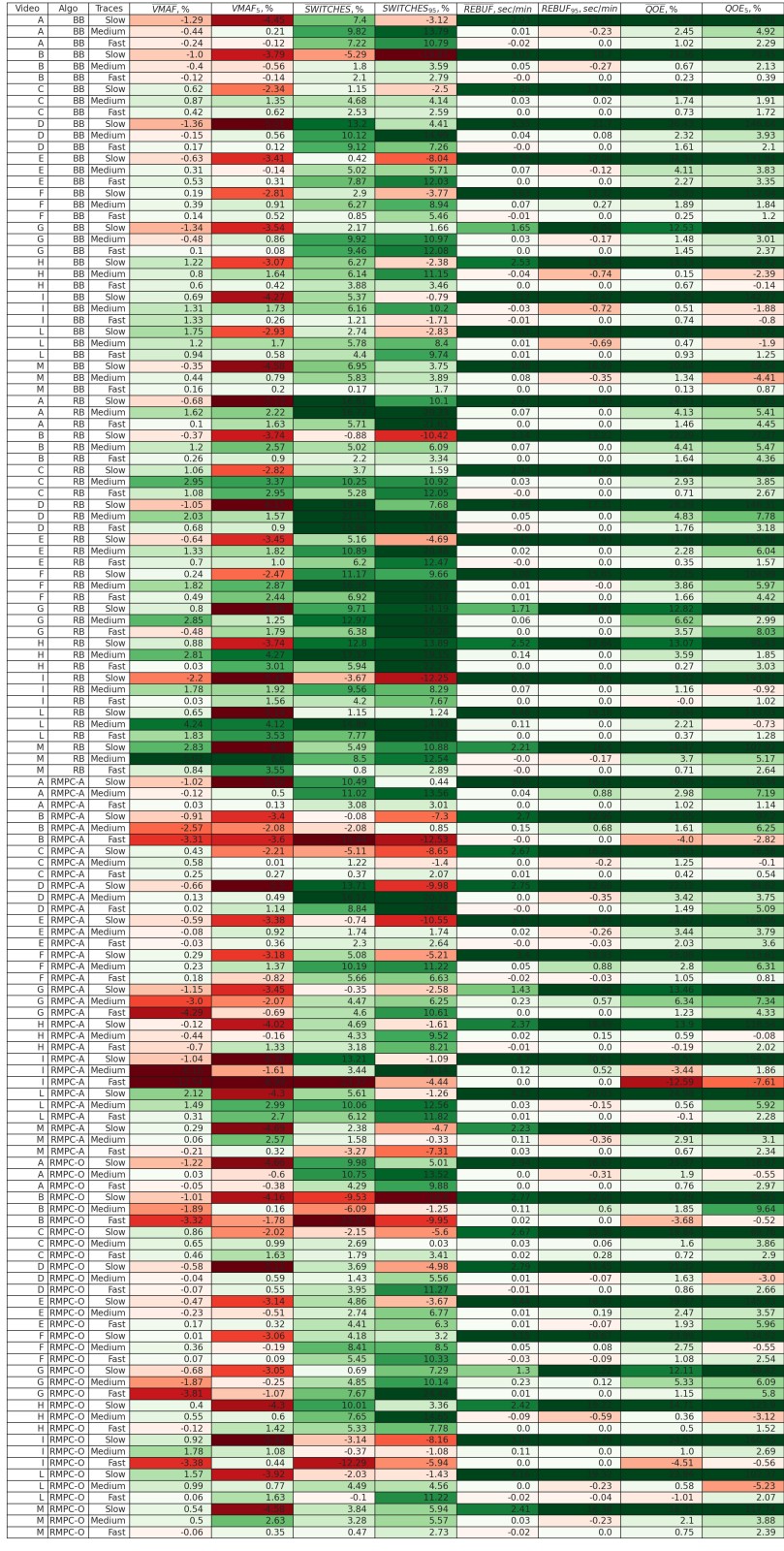

**Fig. 17:** SEGUE*'s results with WideEye segmentation and* $\sigma_{bv}$ *augmentation, compared to Constant. This is a visual summary across 11 videos, 4 algorithms, and 3 trace sets, totaling to ~220 days of streaming time. Each row is one combination of (video, algorithm, trace-bucket). Green colors are improvements, red being deterioration.*

# D    Appendix: Implementation in dash.js

## D.1    Using SEGUE in DASH

To confirm that the simulated results are comparable to real world experiments we run a smaller number of real-time experiments on the DASH JavaScript reference player.

Variable length segments are natively supported by DASH and therefore `dash.js` [7] through the use of a *SegmentTimeline* block in the MPEG-DASH Media Presentation Description (MPD). The *SegmentTimeline* is generated alongside the DASH-compatible media segments using the *sigcues* filter from GPAC [12], which uses the pre-segmented SEGUE chunks and creates the corresponding DASH-playable segments (dashing). For augmented tracks, unavailable chunks are replaced by the corresponding chunks of the standard track during this process, with all the placeholder segments getting removed once dashing is complete.

Per-segment bitrate information is made available to the player using an additional JSON file containing detailed information about all segments. This file is generated as part of the preprocessing step and is based on the simulator input data. During the startup phase of the player, this file is downloaded from a location specified within the MPD.

The `AbrController` of `dash.js` has been adapted to provide per-segment bitrate information supplied by the additional file instead of the average bitrates reported in the MPD. It does so by updating the bitrate (of the next segment) of all tracks to the corresponding values whenever the bitrate list is assembled.

The same approach is used to introduce basic augmentation support: Since tracks cannot be removed or added unless the player switches DASH-`Periods` - which was not a viable option in this case - an unavailable track receives a bitrate of 1 Tbit/s instead. An augmentation-oblivious ABR should not choose a track with such high a bitrate under normal circumstances, while an augmentation-aware ABR can check for this (constant) value to see whether an augmented track is available or not.

The additional information contained in the JSON file can be accessed by an ABR through the `AbrController` if needed, which is used by non-myopic schemes like RMPC to get information about future segments. The three ABRs RB, BB and RMPC have been implemented in JavaScript based on their counterparts used in the simulation.

Three modifications to the default behaviour of `dash.js` were made for our experiments: First, only our custom ABR rule is active, instead of the combination of rules used normally. Second, the start of video playback is intercepted and triggered only once at least 10 seconds of video are in the buffer. Finally, the replacing of already downloaded but not yet played segments ('fast-switching') has been disabled.

## D.2    Experiment setup

The tests run locally on Ubuntu 18.04.5 LTS using Apache webserver (version 2.4.29) to provide the website and video

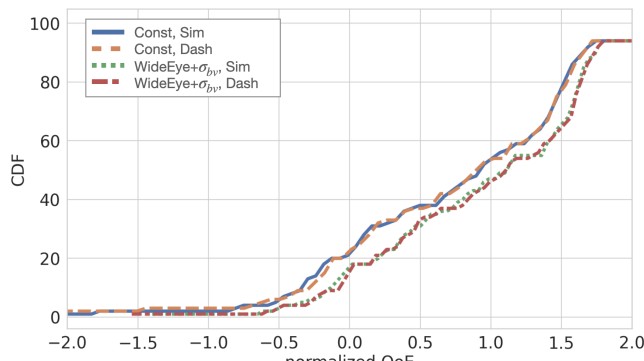

**Fig. 18:** *Comparison of resulting QoE between simulation and* `dash.js` *implementation on the same set of 100 traces for video A and RB. The QoE is normalized by the mean of constant length.*

segments. Selenium WebDriver [6] launches Google Chrome (version 87) in headless mode to load the page and play the video. This process is run from within a Mahimahi shell [24] to emulate different network conditions based on network traces. Metrics from `dash.js` are output through JavaScript log messages, which are retrieved and processed to generate the results.

A set of 100 traces is used to run experiments on video **A** in real-time with `dash.js`, the results of which are then used to compare the simulation results on the same set of traces to. All three ABRs were run on three configurations: constant-length segments without augmentation, the corresponding ABR-specific *WideEye* segmentations without augmentation, and the ABR-specific segmentation with $\sigma_{bv}$ augmentation. The quality metrics are then aggregated using the VMAF 4K model.

## D.3    Results

Overall, results in the QoE distribution are similar. An example of such a comparison is shown in Fig. 18. The figure shows the CDF of the distribution of the QoE for video A and RB, evaluated using VMAF 4K. Lines are plotted for the video without Segue (using constant length segments), and with Segue (WideEye+$\sigma_{bv}$), and each of those once for simulation and execution in DASH.

Similar results are obtained for the ABRs BB and RMPC. In particular, the improvements in performance in the mean of WideEye+$\sigma_{bv}$ with respect to constant are of 5.3%, 6.7% and 5.6% in simulation for BB, RB and RMPC, while in DASH we obtained 5.1%, 7.3% and 5.1%. Improvements have been calculated following the formula expressed in §7.7.

