# OpenReview forum: "Prepare your video for streaming with Segue"
_JSYS/2021/Nov_Papers — Submitted to JSYS Nov 21_

### Official Review · Reviewer_ic3L · 2021-12-13
**Review for Segue**

**Decision:**

Weak reject: interesting papers with flaws, not sure if they can be fixed in three months

**Review:**

**Paper summary**

This paper seeks to optimize video streaming by offline video chunking -- adjusting lengths of video segments (segmentation) and introducing additional quality levels to the bitrate ladder (augmentation) in a manner that improves QoE. The proposed technique, Segue, features the awareness of the video sequence context and the provider-specified ABR algorithm. By accounting for both factors mainly through simulation, Segue shows potential improvement in QoE.

$\newline$

**Strengths**

\+ clear writing with a precise description of the problem and solution

\+ comprehensive evaluation

$\newline$

**Weaknesses**

\- brute-force and simulation-based solution lacks insights and is not thought-provoking

\- missing baselines from prior work in evaluation

$\newline$

**Comments**

Thank you for submitting your work to JSys! This paper has almost impeccable writing and is truly enjoyable to read. The proposed method is easy to understand and appears beneficial to QoE.

That said, I am not so sure about the statements in the paper claiming that it "frames a new direction" or  "sets off a new thread" for ABR streaming and offline chunking. Both aspects that constitute Segue -- segmentation and augmentation -- have been studied before, with the prominent effort originating from Netflix and in the industry as engineering innovations.

Regarding segmentation, Netflix documented a method called "per-shot encode optimization," as mentioned in Section 2.2. However, the authors seem to have misunderstood its mechanism by incorrectly stating that each quality level comprises a number of "fixed-length" segments. In fact, the initial version of their method -- according to Netflix's blog [18] -- chunks a video at boundaries of "shots," each containing a *variable* length of the video. A subsequent [blog](https://netflixtechblog.com/optimized-shot-based-encodes-now-streaming-4b9464204830) further introduced "shot collation" that merges multiple shots into a single chunk, bearing the same spirit as Segue (for a purpose different from QoE optimization though).

As for augmentation, this paper also mentioned the "per-title encode optimization" technique shipped by Netflix years ago, so the only differentiator of Segue on this front is selectively augmenting parts of a video, which sounds like more of a marginal revision than a new "direction" or "thread." In fact, Segue's unique dependence on playback context and ABR algorithm only manifests through basic simulation.

One side effect of positioning itself as a unique direction is that the paper misses baselines from prior work in evaluation.
- Segmentation: Prior work [9, 29, 33] is considered orthogonal to Segue, but what if merging two fragments into a chunk and converting the second keyframe as an inter-frame actually improves the coding efficiency? This will apparently lead to strictly better results (same chunk length with a smaller size) and would reduce (and improve) Segue to the same domain as prior work.
- Augmentation: A naive baseline can be adding one or more additional tracks to the bitrate ladder, which can be used to evaluate Segue's effectiveness and savings (for not encoding and storing the extra tracks). Similarly, although prior work [22, 31] pursues the opposite of Segue, they simply navigate the same tradeoff space by being more generous on the number of tracks, and can also serve as baselines.

Putting the above concern aside, Segue's solution does not seem very novel or inspirational to future work. Brute force and simulation are the most straightforward way that one could approach the outlined problems, not providing many insights to the readers. If Segue could practically deploy and prove its usefulness, the proposed method could be of great engineering value; otherwise, Segue would probably become more ready to publish if it leveraged more interesting insights in its solution, e.g., the interaction between Segue and ABR.

To make my review more suggestive, here is a possible next step to begin with. In Section 4.1, multiple intuitive heuristics have been defined and associated with penalties. Instead of exhaustively enumerating $2^k$ outcomes, can you try the dynamic programming algorithm below off the top of my head? I hope I am not missing something.

Define $P(i)$ as the lowest penalty achieved by segmenting the first $i$ keyframes as chunks, and $C(j, i)$ as the per-chunk cost of merging keyframes $j, j+1, \ldots, i, 1 \le j \le i$ into a chunk -- whether the cost is defined using time, bytes, or time and bytes. Then $P(i) = \min \\{ P(j-1) + C(j, i) \\}, 1\le j \le i$. Given a maximum chunk length, only a small number of $j$s are required to be enumerated and the per-chunk cost table can be pre-computed, so the actual time complexity is close to $O(N)$. Even so, it remains a question as to how segmentation and augmentation could optimize a more sophisticated QoE function given an ABR algorithm without blindly relying on simulation.

Additionally, I am not fully convinced about which segments are more "vulnerable" (besides chunks being overly large): If I were allowed to modify the traces used in the paper arbitrarily, I could choose to increase the instantaneous bandwidth before streaming the "vulnerable" segments such that they are no longer "vulnerable." In other words, bandwidth fluctuation seems to play some fundamental role in offline optimization, but it has not been explored by the paper and could even undermine its findings.

*More comments*

Section 2.2
- As mentioned above, [18] actually uses "variable-length" segments.

Section 3 (above Section 3.1)
- Why is S30 more likely to incur rebuffering than a higher-bitrate chunk S37? Chunks prior to S30 do not look more "challenging" than those before S37. The playback context dependence does not seem to be explained in detail by the paper.

Section 4.2
- Nit: "We describe three augmentation functions": should be "four" and $\lambda_v$ is missing.

**Expertise:**

Actively publishing in this area

**Useful:**

yes

---

### Official Review · Reviewer_RNb5 · 2021-12-15
**A paper that motivates innovations in more flexible ABR track preparation**

**Decision:**

Weak accept: good paper with flaws that can be fixed in three months

**Review:**

### Strengths
* New problem potentially opening another subspace for video streaming research
* Considers both segmentation and augmentation leveraging both heuristics and simulation-based techniques
* Clear paper writing

### Weaknesses
* Improvement numbers are not very large
* Other components beyond chunking are not tuned to best fit variable-length segments and variable numbers of tracks
* Experiments are mostly based on simulation

### Detailed Comments
Thank you for submitting SEGUE to JSys. The argument for adding more flexibility to ABR chunking is backed up by a simple yet effective scheme that carefully segments videos and creates different numbers of versions for different segments. Overall this is a decent paper and I have a few comments as follows.

The idea of having more chunking flexibility is great. Two challenges exist: (1) how to tailor other parts of the video streaming ecosystem (including player logic, ABR algorithm modeling & parameter tuning, QoE function definition & metrics, etc) to make them aware of that and avoid bad interactions, (2) how to optimize chunking given such flexibility. As much as I appreciate the efforts the paper made in addressing the second challenge, I feel there is a lack of detailed justification on the first one, which IMO is even more important in the infancy of this big idea (people will first question this first part too). Without this part being clearly fleshed out, the numbers measured for a specific encoding won’t represent its real performance when everything works in synergy.

SEGUE’s use cases need to be more explicitly specified. Does SEGUE only work for video-on-demand (VoD)? How about live streaming? One challenge for live streaming is that there may not be enough time to run SEGUE and the video complexity information over the entire duration (especially the future scenes) to carefully form tracks. How well would SEGUE work when the ABR algorithm is learning-based, e.g., Pensieve and Fugu? Some platforms randomly choose the ABR algorithm, e.g., Puffer, in this case, whenever an ABR algorithm is chosen, you also need to choose a corresponding ABR track ladder? How do you realize that? Seems that it may need extra control messages in order to inform the server what ABR algorithm the client is currently using.

SEGUE causes extra preparation time overhead when searching for good chunk boundaries for segmentation. Did you evaluate that? This is something Youtube and other open video platforms need to consider as they receive a lot of video uploads every minute and need to encode them ASAP.

Some services can have more than one ABR algo deployed, each for a specific platform (e.g., web v.s. Mobile, or different versions of players), in this case, do you need to prepare another version of tracks for another ABR, which may be too overwhelming?

Need to mention QoE function adjustment early (currently the authors do not mention it until Section 6), otherwise readers will question Fig. 3 in Section 3.2 where the y-axis is QoE yet there is no reasonable definition of QoE for variable-length-segment ABR tracks yet before Section 3.2.

SEGUE's benefits over existing studies (e.g., [24, 28]) are not quantified with experiments, in terms of a better tradeoff between performance and storage overhead.

The benefits of segmentation and augmentation are only evaluated separately.

Network traces are classified into three categories based on mean bandwidth value. Might also be good to do some classification based on bandwidth variability across time.

What if a user starts playing a video from the middle instead of the beginning? Will SEGUE’s prediction cause bad behavior and QoE？

The “Other work” paragraph in the related work section is a bit confusing. Salsify [7] seems not very related, not sure how it relates to SEGUE. It mostly focuses on bitrate/frame-rate adaptation rather than segmentation adaptation. It is also not the first to propose adaptive video conferencing encoding -- WebRTC has been doing that for a long time.

“To avoid the effects of startup behavior, we show results starting at the 25th segment, i.e., 125 seconds into playback.” It may also be worthwhile to look into the first 125s to understand the startup behavior. 125s is not short, and nowadays short videos are the most popular ones so optimizing them considering the startup behavior (instead of “avoiding” it) would be more valuable practically speaking.

Fig. 1: labels are not descriptive enough. Y-labels could use more text to denote the ABR algo and other info. Also fig 1(bcd) could use another set of colors than fig 1a, to avoid establishing wrong mappings at readers’ first glance.

Fig. 2(a): why during 5-10s key frames do not occur frequent despite highly varied bitrates (motions)？

Fig. 3: care to explain why videos A and D behave differently using the same ABR? Also, are these two videos same as video A and D in Table? If so, a forward pointer could be helpful.

Fig. 6: how is VMAF stability defined?

“However, we use this implementation only to verify the fidelity of an orders-of-magnitude faster simulator, which we use for extensive experimentation.” Don’t you also use a real player for video playback in extensive experimentation? Or the extensive experimentation is simulation only?

“Appendix E details the DASH player implementation, demonstrating that it achieves results near-identical to the simulator.” This demonstration was based only on video A, not sure if it generalizes to other videos.

“For the segments highlighted in pink, there is a huge fluctuation in VMAF within the segment boundaries,” true but why do users care? Users don’t even know where the segment boundaries are.

“There can also be value in synchronizing these change points with segmentation, allowing more informed adaptation decisions that account for such changes.” could use some concrete examples to better argue for this.

“RMPC’s implicit assumption that a certain number of segments comprises a long-enough future planning horizon....” I wouldn’t say RMPC assumes that a certain number of segments comprises a long-enough future planning horizon, rather, how to adapt RMPC to a different setting has nothing to do with the RMPC algorithm itself, and the RMPC paper itself never said you should always use 5 segments as lookahead in any situations (e.g., for ABR track ladders with longer/shorter fix-length segments and for those with variable-length segments).

“Adaptive bitrate video streaming comprises two pieces, video encoding, which runs offline at the content provider, and...” May need to mention this offline nature of encoding only applies to VoD.

“Unfortunately, our requests for CAVA’s code were unsuccessful” Why mention this?

“It is unclear to us how or if QoE functions should reward such local improvements.” You may also want to mention you are the first (AFAIK) to put VMAF values into the QoE function so summing over them could be different from summing up bitrate values.

Some of the paper info in Reference (e.g., [10], [11], [14], [25], [31]) are incomplete.

“too many bytes or two few bytes are undesirable”=>”too many bytes or too few bytes are undesirable”

“and their QoE function is used rank the candidate chunkings” => “and their QoE function is used to rank the candidate chunkings”

“providers also want to contain infrastructure costs” => ”providers also want to constrain infrastructure costs”

**Expertise:**

Actively publishing in this area

**Useful:**

yes

---

### Official Review · Reviewer_wqeg · 2021-12-21
**PREPARE YOUR VIDEO FOR STREAMING WITH SEGUE**

**Decision:**

Strong accept: excellent paper that will help the community

**Review:**

Summary:
The paper investigates co-designing encoding and bitrate adaptation for video streaming applications. It argues that existing practice, where videos are segmented into chunks of fixed duration, misses opportunities to optimize QoE further. More specifically, the paper shows that chunks with complex scenes require higher bit rates and are more vulnerable to QoE degradations (e.g., rebuffering). It further argues that using variable-length chunks (segmentation) and more bit rates for complex scenes (augmentation) provides more knobs to optimize user QoE. It formulates the relevant optimization problem and uses simulation-based experiments to quantify QoE improvements for different videos and network conditions.

Thank you for submitting your paper to JSys. I thoroughly enjoyed reading your paper, and I learned a lot in the process. I think the idea of co-designing encoding and adaptation, especially for low-bandwidth networks, is timely and innovative and advances the state-of-the-art in the area.

Pros:
- The idea to co-design encoding and adaptation is well-motivated.  The paper does an excellent job of illustrating why fixed-duration chunks leave a lot desired from a QoE optimization perspective, especially for low-bandwidth networks.
- The evaluation does a good job demonstrating the value of new(ish) QoE optimization knobs.

Cons:
- Though the writing, in general, is good, it is hard to follow at times. For example, the illustrative example in Figure 1 is quite critical to the story. However, it took me a while to grasp the key idea. The section had a few forward references and many moving parts that made it difficult to follow the narrative. To illustrate, the first paragraph in Section 3 says, “Using two rate adaptation algorithms, …” — which two algorithms? Where is the explanation for red/green shades in Figures 1b and 1c? Where can I find an explanation for the difference between 1c and 1d (is there any reference for 1d?)? I am not sure what is a bitrate track here; is it the same as a resolution?
- More elaborate discussion on challenges with the proposed scheme is needed. My first reaction is that the choice of fixed-duration chunks has a lot to do with operational simplicity. NetFlix’s interest in looking beyond naive fixed-duration chunks hints at the operational value of variable-length chunk boundaries. However, the cost of enabling such dynamism in video encoding is still unclear. Where does the proposed approach lie in that design spectrum? The paper does have a cursory discussion on this issue. I will encourage the authors to add a more in-depth discussion on this issue.

**Expertise:**

Actively publishing in this area

**Useful:**

yes

---

### Official Review · Reviewer_UDGa · 2021-12-24
**An interesting paper but needs more**

**Decision:**

Weak reject: interesting papers with flaws, not sure if they can be fixed in three months

**Review:**

Thanks for the interesting and well-written paper! I
appreciated the exploration of the design space of
segmentation and augmentation and the evaluation on real
traces.

The main concern I have with this paper is that the levers
used in this paper, segmentation and selective augmentation,
are proposed without clear reasoning towards why these are
the right solutions to solve the problems discussed in the
prolog of section 3.

Brute-force searching is a nice and simple strategy, but
provides little insight into the structure of the
solution. Specifically, I worry about optimizing encoding
based on ABRs, which are themselves tuned based on encoded
videos. Why not co-design them in the first place, as
suggested in the Discussion section?

In many parts, the methodology feels ad-hoc (setting
parameters, for example). I further feel that the paper
should do more to justify the tradeoffs it achieves (x% more
storage and pre-computation for y% QoE improvement), since
the proposed strategy does not dominate existing
solutions.

There are definitely some clear sparks of ideas in this
paper. However, IMHO the paper could use more thought into
its presentation and reasoning for the advocated
approaches. I hope that the authors could improve these
aspects for their next draft. Please find below more
detailed thoughts on parts of the paper.

intro

Regarding augmentation: I couldn't follow why using more
bitrate tracks for specific scenes (but not the whole video)
is useful. Is the idea to reduce storage at the video
provider?  Does it reduce memory requirements for caching?
Perhaps the authors could quantify how much reduction in
data storage (at the provider, caches, or elsewhere) is
useful, and what SEGUE achieves.

The terms "playback context dependence" is nebulous in the
intro. It would help to define what it means clearly or at
least provide an example how this information is used.

section 2

Industry efforts: Is SEGUE's idea just to capture a subset
of the fixed bitrate tracks? that are available at the video
provider?

"SEGUE’s ideas are orthogonal to this, and address grouping
which scene fragments into seg- ments will result in
beneficial rate adaptation behavior."  I feel the ideas are
indeed related, since the average bitrate of a segment
depends on the homogeneity of the content in the segment.

section 3

"behavior aggregated across a large set of traces". What
traces are these? Which video? which users?

What are the bandwidth ranges corresponding to the slow,
medium, and fast buckets?

One conclusion that a reader could make from plots Fig1(c)
and Fig1(d) is that rate-based algorithms perform better
than buffer-based algorithms for the traces considered in
this paper. (Also, might the problem proposed in the paper
be "solved" if one merely used rate-based adaptation?)
However, this seems to go against the conclusions from
large-scale experiments from the "learning in situ" paper by
Francis Yan and others (NSDI'20), where buffer-based
algorithms outperform others.  Could the authors clarify
this apparent contradiction relative to prior work?
Providing more details on the traces, the environments where
they were obtained, etc. could help dispel possible
misunderstandings earlier.

As background for the discussion in section 3.1, it would
help if the paper clarified whether today's video clients
request a bit rate corresponding to average across all
segments or just the average across the next segment.

"Unfortunately, even with the freedom of variable bitrate
coding, it is sometimes either hard to achieve sufficient
percep- tual quality for complex segments, or
higher-than-necessary bitrate is “wasted” on simple
segments." I couldn't follow this observation. Is the
difference in perceptual quality occurring because of an
implicit assumption that the client will switch to a lower
bitrate (averaged only over the segment, not the entire
video) when the "complex" segments arrive? Why? It's not
obvious that increasing the number of tracks for the complex
segments is the right thing to do here, because that makes
it more possible for the client to experience more changes
in perceptual quality.

What does this phrase mean? "runs of smaller chunks, that
reduce the plan- ning lookahead in terms of time"

section 4

Does adding more tracks for some segments increase the
amount of metadata that must be exchanged during playback
time? Does that have any performance impact especially for
connections with poor bandwidth or high base latency?

section 4.1: It would be useful if the segmentation goal and
strategy are compared to the shot-change detection
algorithms in [18].

How do you choose the target time/bytes?

It was unclear how the "intuitive" segmentation approaches
in section 4.1 help in attaining higher QoE and resolving
the issues pointed out in section 3 regarding poor
segmentation, namely the differences in perceptual quality
across segments or needlessly low quality segments.

Why do segment boundaries from the highest average bitrate
track translate directly to other bitrates? For example,
lower resolutions exhibit lower differences in encoded
bitrates (and, presumably perceptual quality and perceptual
variation between 'complex' and 'simple' segments). Could
they benefit from their own independent segmentation?

Parameters of ABR algorithms may often be tuned given
datasets of videos with existing bitrates.  However,
interestingly, this paper does the opposite, tuning the
available bitrates using existing ABR algorithms. Could you
comment on whether this circularity might push the (ABR +
video preencoding) system into a suboptimal regime? Might it
be useful to use the simulation-based approach to tune both
ABR (parameters) and segmentation simultaneously?

section 4.2: how are thresholds for the various
$\lambda_{b/v/bv}$ chosen?

Regarding simulation for augmentation, what does it mean to
normalize QoE by the bytes overhead? "For each candidate, we
quantify its QoE improve- ment compared to the default
normalized by the overhead in terms of bytes added by that
augmentation sequence. "

section 5

5.1: Could you please provide more details of the QoE
function used in the simulation? Also, please provide some
justification for the threshold values.

section 6

The slow/medium/fast bucketization seem kind of arbitrary.
Why not use logarithmic bucketing covering the entire range?
For example, the fact that 91% of Puffer traces are FAST
seems to indicate that the buckets are highly skewed.

section 7

7.1: It would be great if the paper provides some insight on
why the tradeoff achieved by WideEye is a good one. In
general, it appears that none of the schemes discussed
dominate the naive scheme (constant-time seg) on all
metrics. Could you explain why the tradeoff achieved by the
other schemes is better?

7.3: It could be interesting to design QoE functions that
make significant local improvements stand out in the overall
QoE.

Could the authors comment on the computational cost of
running the simulations to get the noted QoE improvements?
How acceptable might those overheads be in practice, for a
large video provider?


**Expertise:**

Follow the literature closely, last published 5+ years ago

**Useful:**

yes

---

### Meta-Review · Area_Chair_FRdu · 2022-01-06

**Recommendation:** Revise
**Confidence:** 5

**Metareview:**

Thanks for submitting Segue to JSYs. The reviewers feel the paper has investigated an interesting direction with extensive experiments. That said, we feel strongly that the following items must be properly addressed in the revision. And it's important that the authors also clarify/address all other confusions raised in the comments.

Experiments:

1. System overhead [UDGa, wqeg, RNb5, ic3L]: Please evaluate the preparation costs of segmentation and augmentation, in terms of delay and compute overheads (e.g., for a X-minute video with what thresholds, it takes how many minutes on what machines on segmentation and how many minutes on augmentation). Also please evaluate the additional overhead on metafile (the increase of file size and the impact on streaming delay). If the costs are high, please justify them in the context of modern video content providers, especially since the content providers must pay these costs in delay and compute for each ABR algorithm and each video.
2. Segmentation comparison [UDGa, RNb5, ic3L]: Please add a new evaluation to compare the proposed segmentation method with the variable-length segmentation methods studied in [24] (ffmpeg’s built-in variable keyframe insertion based on scene change detection, under different max keyframe intervals), on all videos and test network traces. It’s true that this is a special case of Segue’s segmentation, but the concern is that it’s unclear if Segue will have a substantial improvement or only a marginal one. If it shows a substantial improvement, explain when exactly the proposed method is much (rather than marginally) better. The explanation in 3.2 is based on only one video, and the text is not clear either (see “writing” #1 below). If the improvement is marginal or even negative, please explain why Segue still advances the current practice.
3. Augmentation comparison [ic3L]: Please add a new evaluation to compare the proposed augmentation method with the track filtering schemes in [22,31], on all videos and test network traces. In particular, one can first naively augment each chunk  by adding the average of the bitrates of $s_{i,j}$ and $s_{i,j+1}$, as well as the average for track $j$ for each $j$ (all tracks that might be added by Segue), and then filtering the tracks of each chunk using the techniques in [22,31], with different target quality (VMAF) values. The concern here is that [22,31] is based on the same observations that different tracks can have similar VMAFs and that the same track can have highly variable VMAF values over time, so it is unclear why Segue’s augmentation scheme will have a clear, substantial improvement. If it shows a substantial improvement, please explain when exactly the proposed method is still better despite being based on the same observations. If the improvement is marginal or even negative, please explain why Segue still advances the state-of-the-art.

Writing:

1. [UDGa, ic3L, RNb5] Please downplay the claim that Segue is a first step in a new direction, given that both segmentation and augmentation have been explored extensively in various forms, in both academia and industry. That said, the paper may have something new, e.g., it probably explores the design space more thoroughly, and determines the augmentation and segmentation in a network-trace-aware manner, etc. In both the introduction and the related work sections, the paper should clearly articulate the delta over the existing work [24,22,31] and elaborate on how Segue advances the state-of-the-art. Currently, Section 2.2 is a bit ambiguous/misguided. For instance, [18] doesn’t just use “fixed-length segments”. [22,31] also accounts for the impact of track filtering on rate adaptation algorithms. It’s also unclear why “performance pitfalls” discussed in 3.2 emerge (3.2 neither clearly explains the parameters of the fragmentation scheme, nor does it explain why it has the described performance pitfalls).
2. [ic3L] Elaborate on how segmentation and augmentation decisions may depend on network traces (suggested in detailed comments).
3. [UDGa, RNb5] Discuss how the ABR logic may be changed to accommodate the proposed encoding schemes.
4. [ic3L] Discuss possible ways to reduce the preparation cost and time.
5. [UDGa, wqeg] Explain the traces and the details of the algorithms used in Fig 1. And please improve the presentation of Fig 1 as suggested in detailed comments.
6. [UDGa] "Unfortunately, even with the freedom of variable bitrate coding, it is sometimes either hard to achieve sufficient perceptual quality for complex segments, or higher-than-necessary bitrate is “wasted” on simple segments." It’s hard to see how exactly Fig 1(c) leads to this observation.
7. [UDGa] Add the justification of the various threshold values in the augmentation and segmentation heuristics.
8. Show all rebuffering ratio results of Section 7.1 in graphs.

---

### Decision · Program_Chairs · 2022-01-06

**Decision:**

Revise

**Comment:**

Dear authors,

Based on the reviewers and a subsequent discussion, we have settled on a revised-and-resubmit decision for your paper. The reviews and meta-review will be published later today. As per JSys policy, you have up to three months to submit your revised paper. Please highlight the changes in the revision to facilitate the second round of reviewing.

Once again, we apologize for the delay in communicating our decision.